# Mice and primates use distinct strategies for visual segmentation

**Francisco J Luongo[1†], Lu Liu[1†], Chun Lum Andy Ho[2], Janis K Hesse[1,3,4], Joseph B Wekselblatt[1], Frank F Lanfranchi[1,3,4], Daniel Huber[2], Doris Y Tsao[4,5]***

[1]Division of Biology and Biological Engineering, California Institute of Technology, Pasadena, United States; [2]Department of Basic Neurosciences, University of Geneva, Geneva, Switzerland; [3]Computation and Neural Systems, California Institute of Technology, Pasadena, United States; [4]University of California, Berkeley, Berkeley, United States; [5]Howard Hughes Medical Institute, Berkeley, United States

**Abstract** The rodent visual system has attracted great interest in recent years due to its experimental tractability, but the fundamental mechanisms used by the mouse to represent the visual world remain unclear. In the primate, researchers have argued from both behavioral and neural evidence that a key step in visual representation is 'figure-ground segmentation', the delineation of figures as distinct from backgrounds. To determine if mice also show behavioral and neural signatures of figure-ground segmentation, we trained mice on a figure-ground segmentation task where figures were defined by gratings and naturalistic textures moving counterphase to the background. Unlike primates, mice were severely limited in their ability to segment figure from ground using the opponent motion cue, with segmentation behavior strongly dependent on the specific carrier pattern. Remarkably, when mice were forced to localize naturalistic patterns defined by opponent motion, they adopted a strategy of brute force memorization of texture patterns. In contrast, primates, including humans, macaques, and mouse lemurs, could readily segment figures independent of carrier pattern using the opponent motion cue. Consistent with mouse behavior, neural responses to the same stimuli recorded in mouse visual areas V1, RL, and LM also did not support texture-invariant segmentation of figures using opponent motion. Modeling revealed that the texture dependence of both the mouse's behavior and neural responses could be explained by a feedforward neural network lacking explicit segmentation capabilities. These findings reveal a fundamental limitation in the ability of mice to segment visual objects compared to primates.

**\*For correspondence:**
dortsao@berkeley.edu

[†]These authors contributed equally to this work

**Competing interest:** The authors declare that no competing interests exist.

## Editor's evaluation

There is abundant evidence for differences in the organization of the visual system between primates and rodents. How do these differences yield distinct behaviors in these species? The authors show a major difference in the ability of mice and primates in detecting figures from ground based on motion and texture patterns, revealing a fundamental limitations of mice in segmenting visual scenes.

## Introduction

Primates rely primarily on vision to meaningfully interact with objects in the world. Mice, in contrast, rely far less on their visual system, though they do use visual cues for important behaviors such as hunting, evasion, and navigation (*Evans et al., 2018*; *Fiser et al., 2016*; *Harvey et al., 2012*; *Hoy et al., 2016*; *Leinweber et al., 2017*). The field of mouse vision has attracted great excitement in recent years due to the wealth of tools available for mouse circuit dissection, with many groups adopting the mouse as

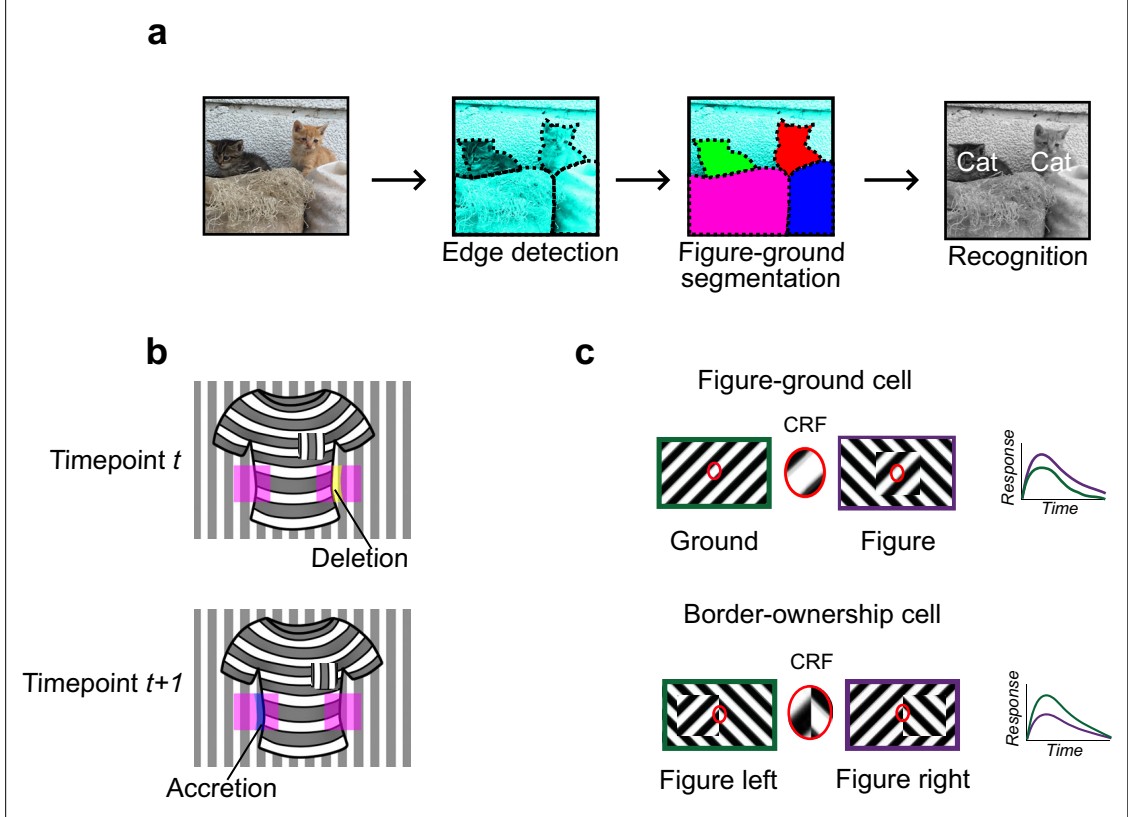

**Figure 1.** Mechanisms for segmentation. (**a**) Schematic representation of a hierarchy for visual perception. Figure-ground segmentation serves as a key intermediate step preceding object recognition. (**b**) Accretion and deletion signals at borders induced by object motion provide a strong cue to distinguish object versus texture edges. As objects move differently from the background, accretion and deletion of parts of the background will occur at object edges, providing local cues for object boundaries and their owners. In contrast to accretion–deletion, texture (e.g., orientation contrast) is locally ambiguous: the pocket does not constitute an object edge, even though it generates a sharp texture discontinuity. (**c**) Top: Figure-ground modulation provides a neural mechanism for explicit segmentation. Here, a hypothetical neuron's firing is selectively enhanced to a stimulus when it is part of a figure (purple) compared to ground (green), even though the stimulus in the classical receptive field remains the same. A population of such neurons would be able to localize image regions corresponding to objects. Bottom: Border-ownership modulation provides an additional neural mechanism for explicit segmentation. Here, a hypothetical neuron's response is modulated by the relative position of a figure relative to an object edge. In the example shown, the neuron prefers presence of a figure on the left (green) as opposed to figure on the right (purple). A population of such neurons would be able to effectively trace the border of an object and assign its owner.

a model for visual perception (*Pinto et al., 2013*) and visually guided decision making (*Abbott et al., 2017*; *Pinto et al., 2019*). Yet the fundamental ethology of mouse vision remains poorly understood. What does the mouse perceive as a visual object?

While work has shown that visual responses in mouse visual cortex share low-level organizing principles with those of primate visual cortex, including temporal/spatial frequency tuning (*Marshel et al., 2011*), orientation selectivity (*Niell and Stryker, 2008*), and contextual surround effects (*Self et al., 2014*; *Keller et al., 2020a*), it remains unclear to what extent the two species share more abstract representations of visual objects and scenes.

In particular, it is unclear whether mice explicitly segment visual scenes into discrete surfaces. Segmentation refers to the identification of borders of each object in a visual scene and assignment of discrete labels to pixels corresponding to each object. In primates, segmentation is a key step in visual processing following early feature extraction (*Frost and Nakayama, 1983*; *He and Nakayama, 1992*; *Lamme, 1995*; *Williford and von der Heydt, 2013*; *Zhou et al., 2000*). For example, in the famous 'face-vase' illusion, human viewers inexorably segment the scene as a face or a vase, with bistable dynamics. A large body of psychophysics suggests that the primate visual system performs segmentation by generating a *surface representation*, an assignment of each retinal pixel to a distinct contiguous surface situated in 3D space (*He and Nakayama, 1992*; *Tsao and Tsao, 2022*; *Figure 1a*).

How could the brain solve visual segmentation? The key visual cue signaling a surface border is a *discontinuity*, an abrupt change in features at the surface border. For example, there is often a change in luminance, orientation, or texture at a surface border. However, this need not be the case: changes in luminance, orientation, and texture can also occur within interior regions of a surface (*Figure 1b*). Conversely, object borders can exist without any change in luminance, orientation, or texture—a fact exploited by animals that use camouflage (*Hall et al., 2013*). Thus, a key challenge of intermediate vision is to identify true object borders using ambiguous local cues. Aiding this goal, there is one cue that is unambiguous: *accretion–deletion*, the appearance or disappearance of pixels forming the background surface due to motion (or binocular disparity) of the foreground surface (*Figure 1b*). Gibson identified accretion–deletion as the *single most important cue to surface organization* because it is unambiguous, invariant to texture, and locally available (*Gibson, 1979*). Psychophysical experiments in humans demonstrate that accretion–deletion alone is able to evoke a vivid percept of an object border (*Nakayama and Shimojo, 1990*). Furthermore, a recent computational theory of surface representation shows how surface segmentation can be computed using local accretion–deletion cues in a simple way without requiring learning, top-down feedback, or object recognition (*Tsao and Tsao, 2022*). Moreover, this new theory shows how such local accretion–deletion cues can be used not only to solve segmentation, but also to solve invariant tracking of objects, widely considered one of the hardest problems in vision (*Pitts and McCulloch, 1947*; *DiCarlo and Cox, 2007*). To summarize, while there are multiple cues to segmentation, accretion–deletion holds a special status for the following reasons: (1) it is unambiguous, occurring only at true object borders and never at internal texture borders; (2) it is cue-invariant; (3) it is especially computationally powerful, supporting not only segmentation but also invariant tracking.

A variety of neural correlates of segmentation have been found in the primate brain. Neurons in primate V1, V2, and V4 modulate their firing according to whether a stimulus is part of the foreground or background (*Lamme, 1995*; *Poort et al., 2012*; *Figure 1c*). Complementing figure-ground signaling, a population of neurons have been found in macaque areas V2 and V4 that explicitly signal object borders. These 'border-ownership' cells respond selectively to figure edges, including those defined purely by accretion–deletion, and are moreover modulated by the side of the figure relative to the edge (*Zhou et al., 2000*; *Qiu and von der Heydt, 2005*; *Figure 1c*).

Behaviorally, mice are capable of texture-based segmentation, in which figure and background are defined by grating patterns with different orientation and/or phase (*Kirchberger et al., 2020*; *Schnabel et al., 2018a*). Consistent with this behavioral capability, cells in mouse V1 show iso-orientation surround suppression (*Self et al., 2014*; *Keller et al., 2020a*) and have been reported to be modulated by figure versus ground (*Schnabel et al., 2018a*; *Kirchberger et al., 2020*; *Keller et al., 2020b*). However, all these studies have used texture-based cues, which are fundamentally ambiguous for solving segmentation (*Figure 1b*). Thus, it remains an open question whether mice are capable of explicit object segmentation, or simply of texture segregation. In contrast, behavioral evidence unequivocally demonstrates that primates possess a mechanism for explicit, cue-invariant segmentation exploiting accretion–deletion (*Nakayama and Shimojo, 1990*).

Here, we take advantage of the ability to record from large numbers of neurons across the mouse cortical visual hierarchy to look for behavioral and neural correlates of visual segmentation in the mouse. We discovered a surprising difference between mouse and human segmentation behavior, which led us to systematically investigate segmentation behavior in three additional species: the macaque, mouse lemur, and treeshrew. We found that the mice and treeshrews, unlike the two primate species, are behaviorally incapable of texture-invariant segmentation. In fact, mice tasked to localize objects with naturalistic textures adopted a strategy of brute force memorization—a cognitively impressive feat. Furthermore, we found no evidence for single neurons in mouse visual cortex modulated by figure/ground or border ownership in a texture-invariant manner. For patterns containing orientation or phase contrast between figure and background, we could decode figure location from population neural recordings, with best decoding in putative ventral stream area LM, followed by RL and V1, but we could not decode figure location for figures with naturalistic texture. A simple feedforward neural network could account for the observed dependence of mouse behavior and neural responses on carrier pattern. Taken together, these findings reveal a fundamental difference between primate and mouse mechanisms for object segmentation, with the mouse relying much more on texture statistics

than the primate. The findings have broad implications for use of the mouse as a model for visual perception.

## Results

### Mice fail to segment objects defined purely by opponent motion

We set out to clarify (1) whether mice are capable of invariantly segmenting figure from ground and (2) whether there exist texture-invariant segmentation-related signals in the mouse brain, as reported in the macaque brain (*Lamme, 1995*; *Zipser et al., 1996*). To address the mouse's ability to segment objects, we designed a two-alternative forced choice task in which mice reported the side of a touch screen that contained a figure for a water reward (*Figure 2a*). We tested mouse segmentation behavior using three classes of stimuli: (1) 'Cross' stimuli, in which the figure consisted of a grating, and the ground consisted of an orthogonal grating; (2) 'Iso' stimuli, in which the figure consisted of a grating, and the ground consisted of a grating at the same orientation, but offset in phase; (3) Naturalistic ('Nat') stimuli, in which both the figure and the ground consisted of 1/f noise patterns (*Figure 2b*, *Video 1*). In each case, figure and ground moved in counterphase providing a differential motion cue with accretion–deletion; this motion cue was essential for defining the figure in the Nat condition. The logic of including these three conditions was as follows: (1) the Cross condition has been used previously in multiple studies of figure-ground segmentation (*Lamme, 1995*; *Poort et al., 2012*) and extra-classical receptive field modulation *Self et al., 2014*; (2) the Iso condition constitutes a slightly more challenging figure-ground segmentation problem due to lack of orientation contrast; nevertheless, the figure can be readily segmented in static images using the phase difference between figure and ground; (iii) the Nat condition allowed us to disambiguate true figure-ground signals from low-level orientation or phase contrast signals.

We first trained mice on four different patterns (two orientations/textures × two sides) for each of the three stimulus conditions (Cross, Iso, Nat, *Figure 2b*). Each session consisted of a single condition (see Methods). The learning curves for the three stimulus conditions were very different (*Figure 2c*). Mice quickly learned the Cross task, reaching 88% performance after 7 days. They were slightly slower to learn the Iso task, reaching 77% performance after 9 days. However, they struggled to effectively learn the Nat task, reaching only around 71% performance after 13 days. We next tested two macaque monkeys on the same task. The monkeys performed at >90% for all three conditions within the first session (*Figure 2d*). Thus, there was a clear difference between the segmentation capabilities of the mouse and primate.

We next wondered whether through more gradual shaping, the mice could learn the Nat task. We trained the mice in a series of stages across 26 training sessions over which the stimulus morphed from the Cross to the Nat condition (*Figure 2e*). For each stage, mice would reach good performance (>80%), followed by a dramatic drop when a new, more difficult stage was introduced. By the end of 26 training sessions, three out of four mice successfully learned to detect the square in the full Nat condition (*Figure 2e*, '100%'). Thus, it appeared that through this gradual shaping, mice had acquired the ability to segment figure from ground using opponent motion.

To confirm this, we next tested the mice on seven new textures. To our surprise, the mice performed near chance on these new textures (*Figure 2f*, mean performance across three mice that had learned the task: 60%; test for significant difference between performance on new textures and performance on last day of noise shaping: p < 0.01 for each mouse, Chi-square test). This lack of ability to generalize suggests that the mice had not in fact learned to segment figure from ground using opponent motion.

How then were they able to perform the Nat task on the trained patterns? We hypothesized that the animals had simply learned to memorize the mapping between the noise patterns in the Nat condition and the appropriate motor response, in effect using a lookup table from four patterns to two actions instead of relying upon visual perception of a figure. If this was the case, we reasoned that we should be able to remove motion from the stimulus and the animals should still perform well. Astonishingly, this turned out to be the case: mice displayed no change in performance upon removal of motion in the stimulus, which completely removed any way of inferring a figure (*Figure 2g*, mean performance across three mice that had learned the task: 87%; test for significant difference between

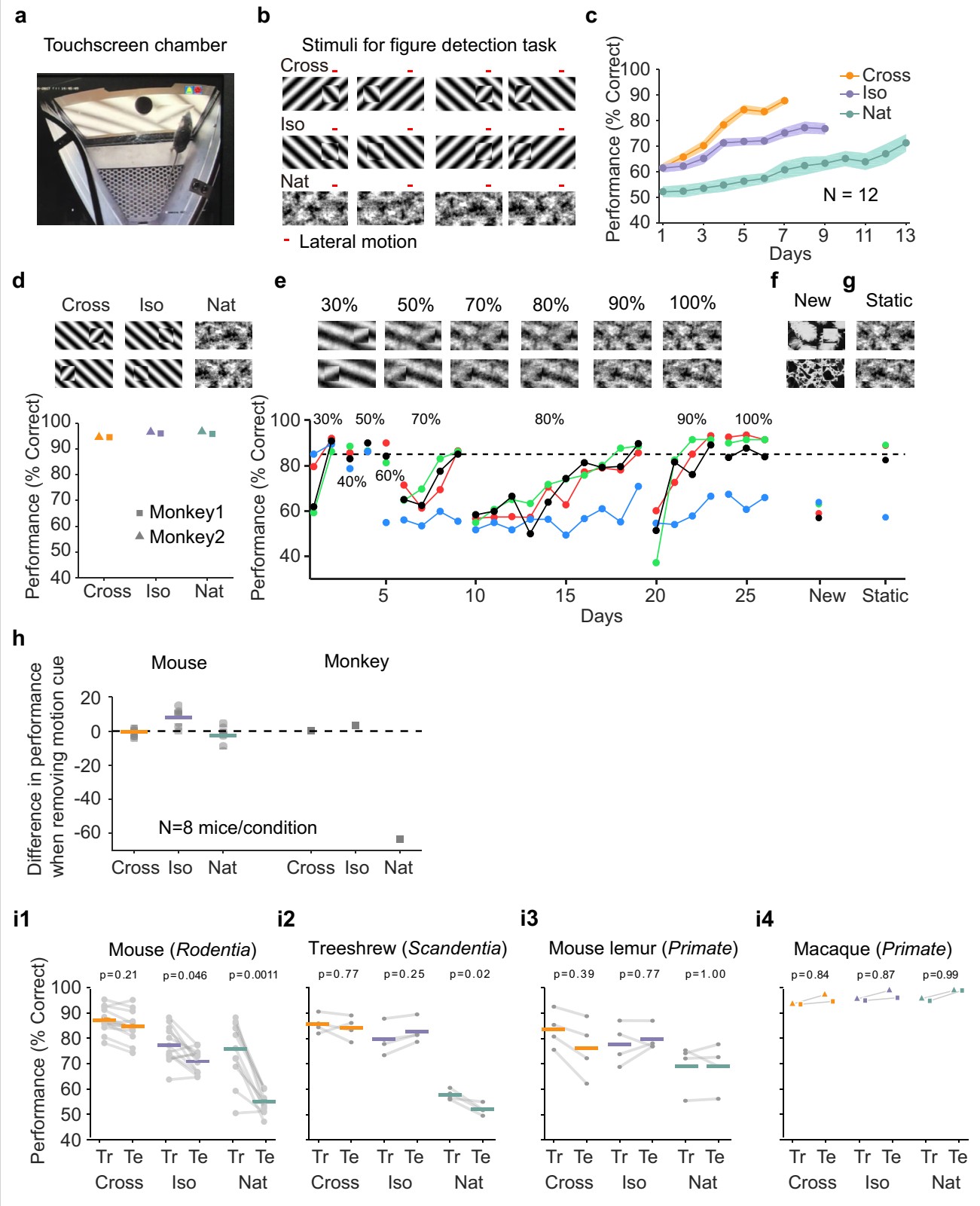

**Figure 2.** Mouse segmentation behavior: mice use orientation contrast but not opponent motion to distinguish figure from ground. (**a**) Mice were trained in a touchscreen paradigm in which they were rewarded for touching the side of the screen containing a texture- and motion-defined figure. (**b**) Mice were tested on three classes of stimuli: 'Cross' where foreground and background patterns consisted of orthogonal gratings, 'Iso' where foreground and background patterns consisted of the same orientation gratings, and 'Nat' where foreground and background patterns consisted of

*Figure 2 continued on next page*

*Figure 2 continued*

naturalistic noise patterns with 1/f spectral content. Initially, four training stimuli were used for each condition. Figure and background oscillated back and forth, out of phase, providing a common opponent motion cue for segmentation across all conditions; the movement range of the figure and background is denoted by the red bar. (**c**) Mean performance curve for 12 mice in the Cross (orange), Iso (violet), and Nat (green) conditions, where the task was to report the side of the screen containing a figure; in each session, one of a bank of four possible stimuli were shown, as in (**b**). Shaded error bars represent standard error of the mean (SEM). (**d**) Performance of two macaque monkeys on the same task. Monkey behavior, unlike that of mice, showed no dependence on the carrier pattern, displaying high performance for all three conditions (Cross, Iso, and Nat). (**e**) Teaching a mouse the Nat condition. Mice that could not learn the Nat version of the task could be shaped to perform well on the task by a gradual training regimen over 20+ days. Using a gradual morph stimulus (see Methods), animals could be slowly transitioned from a well-trained condition (Cross) to eventually perform well on the full Nat task. Each circle represents one mouse. (**f**) Despite high performance on the four stimuli comprising the Nat task, performance dropped when mice were exposed to new unseen textures, suggesting that they had not learned to use opponent motion to perform the task. (**g**) Mice performed just as well on the Nat task even without the opponent motion cue, suggesting that they had adopted a strategy of memorizing a lookup table of textures to actions, rather than performing true segmentation in the Nat condition. (**h**) Left: Change in performance when the motion cue was removed on a random subset of trials. Mice experienced no drop in performance in any of the conditions when static images were displayed instead of dynamic stimuli, indicating they were not using motion information. Note that the static frame was chosen with maximal positional displacement. Right: In contrast, monkeys showed no performance drop in conditions where the figure was obvious in static frames (Cross and Iso), but showed a marked drop in performance for the Nat condition where the figure is not easily resolved without the motion cue. (**i1**) To confirm whether mice used an opponent motion cue in the various conditions, mice (*N* = 12) were trained on an initial set of 4 stimuli (Tr; 2 sides × 2 patterns/orientations, as in (**b**)). After performance had plateaued, they were switched to 10 novel test conditions (Te; 2 sides × 5 patterns/orientations). Animals mostly generalized for Cross and Iso conditions but failed to generalize for the Nat condition (p = 0.0011, ranksum test), suggesting they were unable to use the opponent motion in the stimulus. (**i2**) Same as (**i1**) for treeshrews. Like mice, treeshrews failed to generalize for the Nat condition (p = 0.02, ranksum test). In contrast, two primate species: mouse lemurs (**i3**) and macaques (**i4**) were able to generalize in the Nat condition, suggesting they were able to use the opponent motion cue in the task (p = 1.00 mouse lemur, p = 0.99 macaque; ranksum test).

The online version of this article includes the following figure supplement(s) for figure 2:

**Figure supplement 1.** Natural texture task shows advantage for learning figures with cross-oriented energy.

**Figure supplement 2.** Additional behavioral performance statistics.

---

performance on static textures and performance on last day of noise shaping: p > 0.01 for each mouse, Chi-square test).

We next tested all three conditions (Cross/Iso/Nat) in separate cohorts of mice to further examine whether the animals were indeed discarding the opponent motion cue. We tested them on a static condition of the task after training on the motion task. As before, these mice showed no drop in behavioral performance for any of the three conditions (*Figure 2h*, Cross: p = 0.67, Iso: p = 0.02 (increasing), Nat: p = 0.26), confirming that the animals were not using the motion cue for figure detection. While this was not surprising for the Cross and Iso cases, as the single static frame still had strong edge contrast due to orientation/phase differences and thus contained a clear figure that could be detected, it was surprising for the Nat condition which had minimal cues for the figure when static. Thus, this experiment further confirmed that mice did not use the opponent motion cue to perform the segmentation task.

For comparison, we performed the same test on two monkeys. Their performance showed a very different pattern. Like mice, monkeys did not display a drop in performance in the Cross and Iso conditions. For the Nat condition, however, monkeys showed a dramatic drop in performance when motion cues were removed from the stimulus (*Figure 2h*, Fisher's exact test, p-vals, Monkey 1: Cross: 1.0, Iso: 0.41, Nat: 9.7e−14; Monkey 2: Cross: 0.77, Iso: 0.75, Nat: 2.2e−19). This experiment reveals that monkeys and mice used fundamentally different strategies to solve the Nat condition: monkeys used the opponent motion cue, resulting in a dramatic drop in performance upon removal of motion, while mice used a learned lookup table mapping patterns to responses.

Given the inability of mice to generalize to new textures for the Nat condition (*Figure 2f*), we wondered whether the same would hold true for the Cross and Iso conditions. We next trained 8

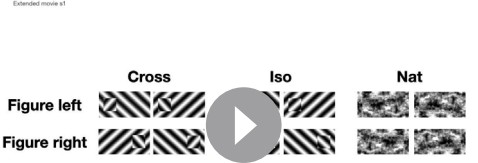

**Video 1.** Examples of dynamic stimuli used to test mouse segmentation behavior. Left to right: Cross, Iso, and Nat conditions.

https://elifesciences.org/articles/74394/figures#video1

new animals on the Cross and Iso tasks with 4 patterns (2 orientations × 2 positions), and then tested their ability to generalize to 10 new patterns from the same class (see Methods). We found that mice were able to generalize well for the Cross condition (*Figure 2i1*, left, mean performance drop = 2.47%), and moderately well for the Iso condition (mean performance drop = 6.55%). However, for the Nat condition, performance dropped almost to chance (mean performance drop = 20.61%; p = 0.0011, ranksum test), consistent with the result of our previous experiment (*Figure 2f*). One obvious concern is that the mice simply could not see the motion cue in the Nat condition due to their low visual acuity; this concern is addressed in detail further below.

Since mice showed their best generalization performance for cross-oriented gratings, this suggested that orientation contrast is a key feature used by mice to solve the task of localizing the figure. This in turn suggested that we might improve the animal's performance on random textures by introducing an element of orthogonal orientation. To test this, we generated two new sets of figure-ground stimuli starting from random textures: (1) 'Iso-tex' stimuli, in which a square was cropped from the texture and placed in either the left or right position, and (2) 'Cross-tex' stimuli, in which the same square was rotated 90°, increasing orientation contrast (*Figure 2—figure supplement 1a*); for both sets of stimuli, opponent motion between figure and ground was added. To compare the generalization ability of mice on these two classes of stimuli, we first measured baseline performance on Iso-tex and Cross-tex stimuli drawn from seven random textures. We then trained mice on Iso-tex and Cross-tex stimuli drawn from a different set of 30 textures. Finally, we re-measured performance on the original set of seven textures (*Figure 2—figure supplement 1b*). While there was no difference in baseline performance between the two conditions, a significant difference emerged during training (*Figure 2—figure supplement 1c–f*). Animals trained on the Iso-tex condition achieved a mean performance of 57% after 14 days of training (*Figure 2—figure supplement 1c, d, g*), whereas animals trained on the Cross-tex condition achieved 67% correct after 14 days (*Figure 2—figure supplement 1e–g*)**,** indicating that a strong orthogonal component could aid the mice in performing the task. However, despite above chance performance on the bank of 30 random textures, just as before, mice were largely unable to utilize any information about the motion cue, as demonstrated by their drop back to initial performance for the original bank of seven textures (*Figure 2—figure supplement 1d, f, h*). Overall, our behavioral results suggest that mice adopt a strategy for object localization that relies heavily on orientation contrast and phase differences between figure and ground and is blind to opponent motion cues.

## Comparing segmentation behavior in mouse, macaque, treeshrew, and mouse lemur

The striking difference between mouse and macaque segmentation behavior inspired us to run the generalization test of *Figure 2i1* on two macaque monkeys (*Figure 2i4*). The macaques showed a very different behavioral pattern compared to the mice: they were able to generalize to unseen patterns for all three conditions, indicating that they were capable of performing segmentation using the opponent motion cue, and had not simply memorized the texture pattern in the Nat condition like the mice.

This clear difference between the behavioral strategies for visual segmentation used by mice versus macaques further inspired us to perform the same pattern generalization test (i.e., train on one set of patterns/orientations, test on a different set of unseen patterns/orientations) in two additional species: (1) a second mammalian species of the order scadentia (*Tupaia belangeri*; treeshrew), and (2) a second primate species (*Microcebus murinus*; mouse lemur). The treeshrews performed similar to mice, displaying generalization for the Cross and Iso conditions but not the Nat condition (*Figure 2i2*). In contrast, and similar to macaques, the mouse lemurs were readily able to generalize for all three conditions (*Figure 2i3*), implying that they, like the macaques (*Figure 2i4*), were able to perform visual segmentation using the opponent motion cue. Training curves for all four species on this task are shown in *Figure 2—figure supplement 2a, b*. Taken together, these results provide strong evidence that primates including mouse lemurs, macaques, and humans all use a visual segmentation strategy exploiting opponent motion cues, in contrast to mice and treeshrews, which rely on texture cues to perform visual segmentation and are incapable of using opponent motion cues.

## Controlling for visual acuity

The peak acuities of the four species studied vary over two orders of magnitude: 5 cpd (macaque fovea, *Merigan et al., 1991*), 0.5 cpd (mouse lemur, *Ho et al., 2021*), 0.5 cpd (treeshrew, *Petry et al., 1984*), and 0.05 cpd (mouse, *Prusky et al., 2000*). Thus, one obvious concern is that mice and treeshrews could not detect the square in the Nat condition due to their lower visual acuity compared to macaques. Several pieces of evidence argue against this. First, mouse and treeshrew visual acuity differ by an order of magnitude, yet neither species could perform the task. Second, even more compellingly, the treeshrew and mouse lemur have *identical visual acuity*, yet the mouse lemur could perform the task while the treeshrew could not. Third, mouse lemurs, treeshrews, and mice all performed the task in a freely moving paradigm that naturally generates a parametric range of spatial frequencies and figures sizes on the animal's retina (for the mouse, the size of the behavior box was 25 cm; this generates a variation in figure size from 15° to 143° assuming the mouse position varied from 23 to 1 cm from the screen; and a tenfold variation in spatial frequency). Despite this, mice and treeshrews still could not perform the task. Importantly, 23–52% of the stimulus energy in the Nat stimulus had frequency less than 0.064 cpd, the peak contrast sensitivity of the mouse (*Prusky et al., 2000*), when the mouse position varied from 23 to 1 cm from the screen. Fourth, the fact that the mouse could learn to distinguish the Nat patterns after extensive training (*Figure 2c*) directly demonstrates that *the mouse could perceive the spatial frequencies present in the Nat stimulus*. Finally, one might be concerned that the amplitude of the counterphase motion (14 pixels) was too small for the mouse to perceive. To control for this, we trained the same eight mice as in *Figure 2i1* to perform the Nat condition in a modified situation in which the background was static. The animals learned this modified Nat task much more quickly, indicating that they could see the motion cue (*Figure 2—figure supplement 2c*); their deficit was specific to seeing the figure defined by opponent motion. Together, all these pieces of evidence argue strongly that the differences we observed between mouse/treeshrew on one hand, and macaque/mouse lemur on the other, cannot be attributed to differences in visual acuity.

## Absence of texture-invariant figure signals in mouse visual cortex

Given the evident inability of mice to perform texture-invariant visual segmentation, a natural question arises: what segmentation-related signals are available in mouse visual cortex to decode the location and boundary of an object? To address this, we recorded responses in mouse visual cortex to figure-ground stimuli defined by both texture and opponent motion using (1) electrophysiology with a 64-channel silicon probe, and (2) 2-photon calcium imaging. This stimulus was essentially identical to the one used to test mouse behavior, except the figure location changed from trial to trial in order to cover the cells' receptive fields. We compared responses across three distinct mouse visual areas: primary visual cortex (V1), a putative ventral stream area (LM), and a putative dorsal stream area (RL) (*Wang et al., 2011*).

We first localized visual areas using wide-field imaging in GCAMP6s transgenic animals and used vasculature maps to guide subsequent two photon and electrophysiology experiments (*Figure 3a, b*) (see Methods). We then measured the receptive field centers of neurons using either a sparse noise stimulus or a spatially isolated flashing Gabor of varying orientations. Imaging and electrophysiology data were generally consistent. For most analyses below (with the exception of *Figure 3f, g* and Figure 6d, f), we present electrophysiology data, as it had better temporal resolution and gave better receptive field estimates (due to absence of neuropil activity leading to blurring of receptive fields). The latter was critical as the analyses depended on accuracy of receptive field estimates for single cells.

To visualize a neuron's response to figure, ground, and borders, we computed a 'figure map' consisting of the neuron's mean response to a figure centered at each of 128 (16 × 8) positions across the visual field (*Figure 3c*, *Video 2*). The square figure appeared for 250 ms at each position. The stimulus closely mimicked that used for behavioral tests; in particular, the square moved in counterphase to the background. This stimulus enabled us to measure responses of each neuron to figure, ground, and borders, as on any given trial a particular location contained figure, ground, or borders. For example, the figure map of an ideal figure cell would reveal a square corresponding to the figure (*Figure 3d*, left), that of a border cell would reveal stripes corresponding to the figure borders matching the orientation of the cell (*Figure 3d*, middle), and that of an ON-cell would reveal phase-dependent

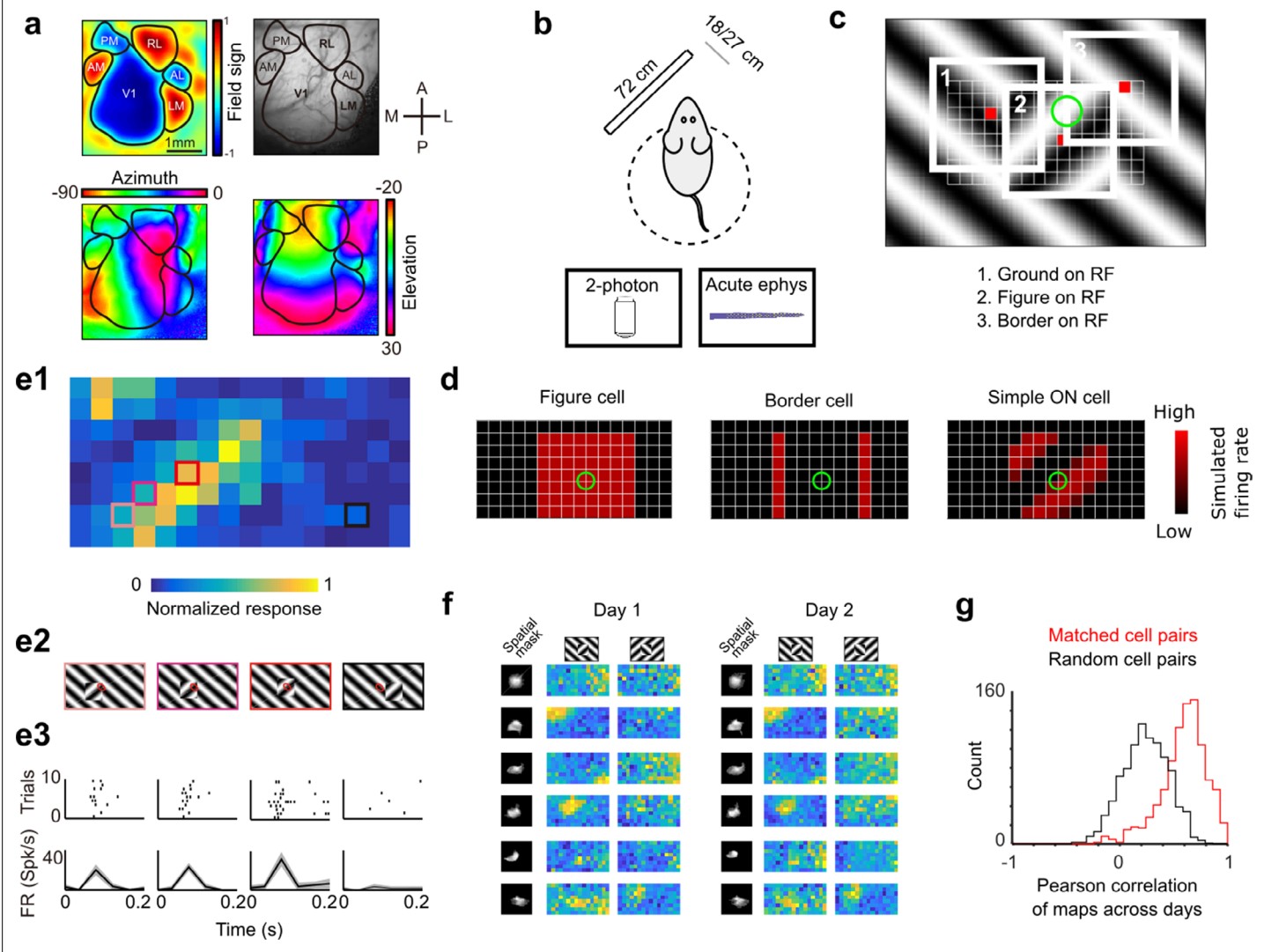

**Figure 3.** Approach for measuring neural correlates of segmentation-related modulation in mouse visual cortex. (**a**) Widefield imaging of GCaMP transgenic animals was used to localize visual areas prior to neural recording. A drifting bar stimulus was used to compute azimuth (bottom-left) and elevation (bottom-right) maps across of the visual cortex. From these maps, a field-sign map (top-left) was computed, allowing delineation of cortical visual areas (top-right). Alignment to vasculature maps guided subsequent electrophysiology or two-photon recordings to V1, LM, or RL. (**b**) Rodents were allowed to run freely on a spherical treadmill, with a 72-cm width (32-inch diagonal) screen centered either 18 or 27 cm away from the left eye, while undergoing either electrophysiology or two-photon imaging. (**c**) The stimulus consisted of a texture- and motion-defined square that was flashed randomly across a grid of 16 horizontal positions × 8 vertical positions (128 positions total). On any given trial, a neuron with a given receptive field location (schematized by the green circle) was stimulated by (1) ground, (2) figure, or (3) edge, enabling us to measure both figure-ground and border-ownership modulation. (**d**) Schematic response maps to the stimulus in (**c**). Left: A 'figure cell' responds only when a part of a figure is over the receptive field. Middle: A 'border cell' responds only when a figure border falls on the receptive field and has orientation matching that of the cell (here assumed to be vertical). Right: A simple cell with an ON subunit responds to the figure with phase dependence. (**e1**) Mean response at each of the 128 figure positions for an example V1 cell. Colored boxes correspond to conditions shown in (**e2**). (**e2**) Four stimulus conditions outlined in (**e1**), ranging from receptive field on the figure (left) to receptive field on the background (right). (**e3**) Raster (top) of spiking responses over 10 trials of each stimulus configuration and mean firing rate (bottom). Error bars represent standard error of the mean (SEM). (**f**) Example response maps from V1 using two-photon calcium imaging show reliable responses from the same neurons on successive days. Shown are six example neurons imaged across 2 days. Neurons were matched according to a procedure described in the Methods. Colormap same as in (**e1**). Spatial masks are from suite2P spatial filters and are meant to illustrate qualitatively similar morphology in matched neurons across days. (Correlation between days 1 and 2: first column: 0.46, 0.90, 0.83, 0.82, 0.58, 0.36; second column: 0.39, 0.96, 0.93, 0.66, 0.85, 0.47.) (**g**) Distribution of Pearson correlations between figure maps for all matched cell pairs (red) and a set of randomly shuffled cell pairs (black). Neurons displayed highly reliable responses to the stimulus ($N$ = 950 cell pairs in each group, mean = 0.5579 for matched vs. mean = 0.2054 for unmatched, p = 1e−163, KS test, $N$ = 475 cells matched, day 1: 613/day 2: 585).

**Video 2.** Examples of dynamic stimuli used to compute figure maps for cells in electrophysiology and imaging experiments. Left to right: Cross, Iso, and Nat conditions.

https://elifesciences.org/articles/74394/figures#video2

responses to the figure (*Figure 3d*, right). As these model units illustrate, the figure map is a function of a cell's receptive field location, low-level stimulus preferences (e.g., orientation selectivity, contrast polarity selectivity), and high-level stimulus preferences (figure/ground selectivity, border selectivity). Thus, the figure map yields a rich fingerprint of a cell's visual selectivity.

*Figure 3e1* shows the figure map for one example cell from V1 recorded with electrophysiology in the Cross condition. Responses to a subset of four stimuli (*Figure 3e2*) revealed a decreasing response as the figure moved off the receptive field (*Figure 3e1–e3*). We confirmed that these figure maps were highly stable using two-photon imaging across multiple days. *Figure 3f* shows figure response maps obtained from six example cells across two different days. The mean correlation between maps from matched cell pairs across different days was very high (*N* = 950 cell pairs, Pearson *r* = 0.5579 matched vs. Pearson *r* = 0.2054 unmatched, KS test p = 1e−163, *Figure 3g*).

To fully characterize responses of neurons to figure, ground, and borders, we obtained figure maps using the same three conditions as those used earlier in behavior (Cross, Iso, and Nat) (*Figure 4a*). We presented two orientations/textures for each of the three conditions. In V1, we often found cells that showed orthogonal stripes for the two different cross patterns (*Figure 4b*), as expected for an ON- or OFF-cell (*Figure 3d*, right). We failed to find any cells in V1, LM, or RL that showed consistent figure maps across the different conditions (*Figure 4a–d*). To quantify this across the population, we computed distributions of the mean Pearson correlation between figure maps across all possible pairs from the six conditions: the values were centered around 0 (V1: 0.032, LM: 0.034, RL: 0.015) (*Figure 5a*). Within each condition, the mean Pearson correlation between figure maps was also centered around 0 (*Figure 5b*). This shows that across the population, selectivity to figure location within individual neurons was strongly dependent on the specific texture of the figure.

We next quantified selectivity for figure ground and border ownership, two of the hallmark segmentation-related signals reported in the macaque visual system, across the V1, LM, and RL cell populations. We only analyzed neurons that had significant receptive field fits (see Methods); furthermore, we confined our analysis to neurons with receptive fields centered within four degrees of the monitor center, to ensure that there were an adequate number of figure, ground, and left/right trials from which to compute the neuron's modulation indices. For each of the three conditions, we defined a figure-ground modulation (FGM) index as $FGM = \frac{(R_{Fig} - R_{Back})}{(R_{Fig} + R_{Back})}$, where $R_{Fig}$ is the mean response across the two patterns for the condition within the figure zone, i.e., the 2 × 2 (6° × 6°) grid of locations centered on the monitor (4 locations × 10 trials = 40 figure trials) and $R_{Back}$ is the mean response across the two patterns for the condition in the background zone, that is, the leftmost and rightmost column of locations (2 × 8 locations × 10 trials = 160 background trials) (*Figure 5c*). We were extremely conservative in our selection of figure and ground locations to avoid any mistakes in labeling due to uncertainties in receptive field location. Distributions of FGM indices were approximately centered on zero, with a slight rightward shift for Cross (Cross: 0.07, Iso: −0.014, Nat: −0.03) (*Figure 5d*). To determine significance of FGM indices, for each cell and condition, we estimated a bootstrapped p value for the corresponding FGM value using distributions of shuffled trials (see Methods); we found no neurons that showed significant figure modulation across more than three conditions (*Figure 5e*).

We quantified border-ownership selectivity in a similar way. We defined a border-ownership modulation (BOM) index as $BOM = \frac{(R_{Left\ border} - R_{Right\ border})}{(R_{Left\ border} + R_{Right\ border})}$, where $R_{Left\ border}$ is the mean response across the two patterns for the condition within the left border zone (Column 4; 8 locations × 10 trials = 80 left edge trials), and $R_{Right\ border}$ is the mean response across the two patterns for the condition within the right border zone (Column 12; 8 locations × 10 trials = 80 right edge trials) (*Figure 5f*). Distributions of BOM indices were approximately centered on zero (Cross: −0.02, Iso: −0.01, Nat: −0.07) (*Figure 5g*). We found no neurons that showed significant BOM across more than three conditions

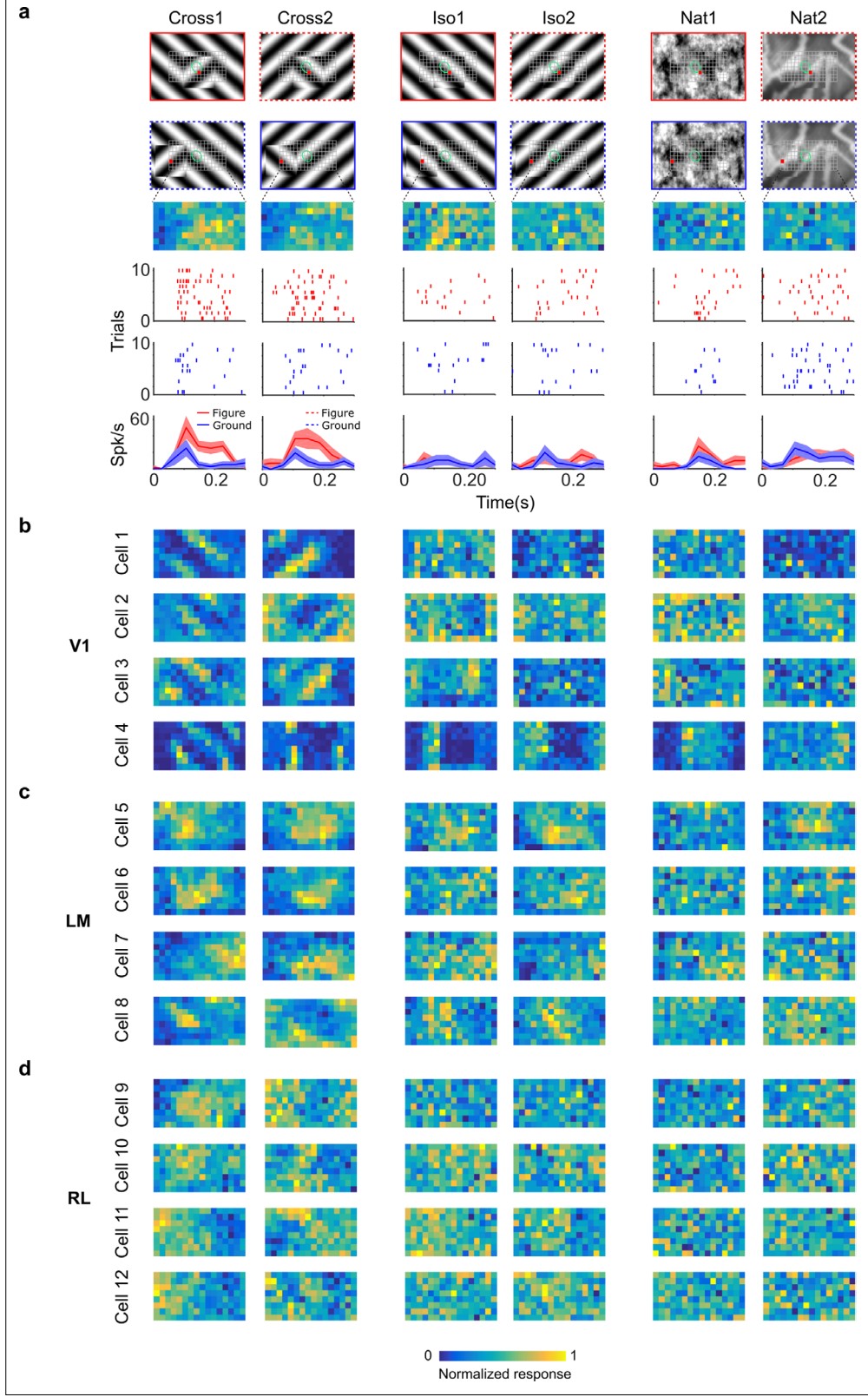

**Figure 4.** Segmentation-related modulation across mouse visual cortex is pattern dependent. (**a**) To search for segmentation-related neural signals, we adapted the stimulus shown in *Figure 3c* (a figure switching between 128 positions) to the three conditions that we had tested behaviorally (Cross, Iso, and Nat). As in the behavior experiments, for each condition we presented two variants (different orientations or patterns). Rows 1 and 2: Two

*Figure 4 continued on next page*

*Figure 4 continued*

example frames (with square at different positions) are shown for each of the six stimulus variants. Overlaid on these example stimuli are grids representing the 128 possible figure positions and a green ellipse representing the ON receptive field. Note that this receptive field is the Gaussian fit from the sparse noise experiment. Row 3: Example figure map from one cell obtained for the conditions shown above. Rows 4 and 5: Example rasters when the figure was centered on (red) or off (blue) the receptive field. Row 6: PSTHs corresponding to the rasters; shaded error bars represent standard error of the mean (SEM). (**b**) Figure maps for each of the six stimulus variants for four example neurons from V1 (responses measured using electrophysiology). Please note that for all of these experiments the population receptive field was centered on the grid of positions. (**c**) Figure maps for each of the six stimulus variants for four example neurons from LM. (**d**) Figure maps for each of the six stimulus variants for four example neurons from RL.

(*Figure 5h*). Thus overall, we found weak signals for FGM and BOM, which moreover depended strongly on specific texture condition, across the three mouse visual areas surveyed.

Mean time courses of responses across the population to figure, ground, and border confirmed the strong texture dependence of FGM and BOM signals (*Figure 5i–k*). While there was clear enhancement in response to the figure/border versus ground for the Cross condition starting at the earliest time point of response (*Figure 5i*), differences were much smaller for Iso and Nat conditions (*Figure 5j, k*). Comparison of time courses across areas revealed a more distinct response difference between figure and ground conditions in LM compared to V1, and V1 compared to RL, with strong texture dependence of segmentation signals in all three areas (*Figure 5—figure supplement 1*).

## Neural decoding of figure position mirrors behavioral performance

The neural data so far shows a clear lack of texture-invariant segmentation signals in mouse visual areas V1, LM, and RL (*Figures 4 and 5*). This is consistent with the mouse's inability to generalize segmentation across textures for the Nat condition (*Figure 2e, f, i1*). However, the mouse was behaviorally able to generalize segmentation in the Cross and (to a lesser extent) Iso conditions (*Figure 2i1*). To what extent can the neural signals in mouse visual cortex explain this pattern of behavior?

To address this, we quantified how well we could read out the position of a figure on a given trial using a linear decoder of neural responses. Within a single condition (Cross, Iso, and Nat), decoding position would be trivial for a single stimulus pattern: a set of ON cells with localized receptive fields like the hypothetical unit in *Figure 3d* (right) would be able to solve this task, as long as cells respond reliably and differentially to stimuli with different figure locations. How well could a decoder trained on the mouse's neural responses *generalize* segmentation across textures within a class? For each of the three conditions, we pooled trials for the two orientations/patterns, and then trained a least squares linear regression model using 50/50 cross-validation over trials (*Figure 6a*). *Figure 6b* shows decoded versus actual figure position for varying numbers of cells; for convenience, we decoded azimuth position. Decoding improved monotonically with the number of cells used.

We quantified decoding performance as the variance in the azimuth position explained by the linear model (*Figure 6c*). Using electrophysiology data, we found that on average neural decoding was best for the Cross condition ($r^2 = 0.89$ for 200 cells), followed by Iso ($r^2 = 0.53$ for 200 cells), and then Nat ($r^2 = 0.09$ for 200 cells). This dependence of position decoding on texture condition (Cross > Iso > Nat) matched the ranking observed in the behavioral performance of animals on the generalization stage of the figure localization task (*Figure 2i1*). In particular, variance explained was close to zero for the Nat condition.

Using imaging data, we found the same qualitative pattern, though overall decoding performance was worse than that obtained from electrophysiology data for the same number of neurons (*Figure 6d*), likely due to the fact that the calcium signal is significantly noisier (*Berens et al., 2018*).

We next examined decoding performance for each of the conditions as a function of visual area. For both the Cross and Iso conditions, decoding was best for LM followed by V1 and RL (for $N = 120$ cells: Cross: LM > RL: $p < 10^{-4}$, LM > V1: $p < 10^{-4}$, V1 > RL: n.s.; Iso: LM > RL: $p < 10^{-4}$, LM > V1: $p < 10^{-4}$, V1 > RL: $p < 10^{-4}$, rank sum test) (*Figure 6e*). A similar relationship was observed with imaging data (*Figure 6f*), albeit with better decoding for RL compared to V1 for Cross.

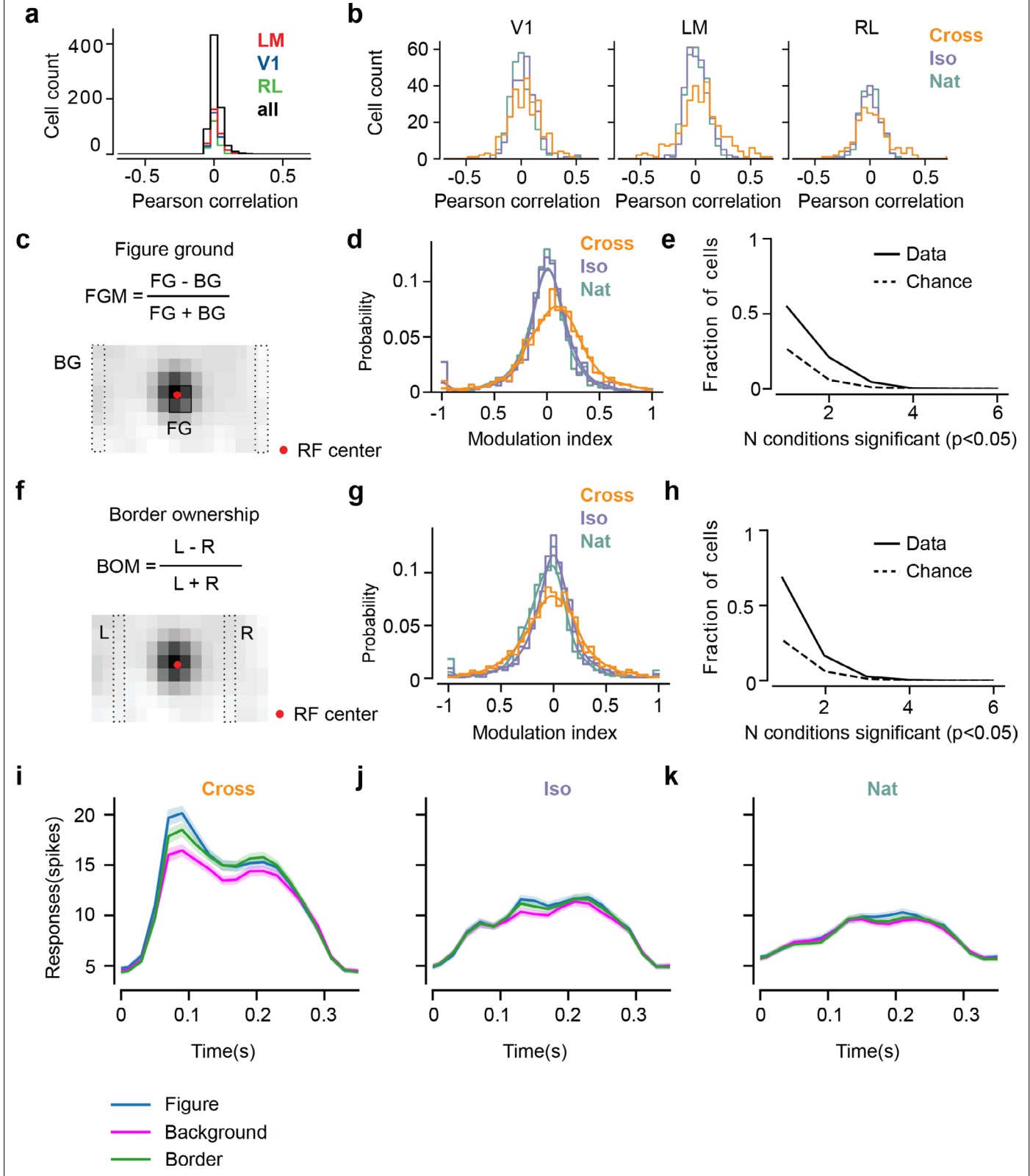

**Figure 5.** Mice lack consistent segmentation signals across texture conditions. (**a**) Distribution of Pearson correlations between figure maps across all $\binom{6}{2}$ pairs of conditions. No neuron in any area showed high correspondence (signified by non-zero mean) across all conditions tested, indicative of a texture-invariant figure response. (**b**) Distribution of Pearson correlations between figure maps across the two stimulus variants for each condition

*Figure 5 continued*

(orange: Cross, violet: Iso, green: Nat) and across visual areas (left: V1, middle: LM, right: RL), V1 = 260 cells, LM = 298 cells, RL = 178 cells. Means and p value testing for difference from 0 for each condition and area: 0.03, 9.6e−4 (V1, Cross), 0.06, 1.9e−6 (LM, Cross), 0.02, 1.8 (RL, Cross), 0.03, 3.1e−5 (V1, Iso), 0.02, 2.1e−2 (LM, Iso), −0.0038, 0.60 (RL, Iso), 0.006, 0.92 (V1, Nat), 0.005, 0.44 (LM, Nat), 0.009, 0.21 (RL, Nat). (**c**) A figure-ground modulation (FGM) index was computed by taking the mean response on background trials (positions outlined by dashed lines) and the mean response on figure trials (positions outlined by solid line) and computing a normalized difference score. Note, the black/white colormap in this figure corresponds to a model schematic RF. (**d**) Distribution (shown as a kernel density estimate) of FGM indices for Cross (orange), Iso (violet), and Nat (green) conditions, pooling cells from V1, LM, and RL. (**e**) Fraction of cells with FGM index significantly different from zero (p < 0.05) for *N* stimulus variants (out of the six illustrated in *Figure 4a*). Dotted gray line represents chance level false-positive rate at p < 0.05 after six comparisons. For this analysis, FGM was computed similarly as (**d**), but responses were not averaged across orientations/patterns within each condition; thus each cell contributed 6 FGM values. (**f**) A border-ownership modulation index was computed by taking the mean response on left edge trials (positions outlined by dashed rectangle marked 'L') and the mean response on right edge trials (positions outlined by dashed rectangle marked 'R') and computing a normalized difference score. (**g**, **h**) Same as (**d**), (**e**), but for border-ownership modulation indices. Mean response time courses across all cells in V1, RL, and LM to figure, ground, and border in the Cross (**i**), Iso (**j**), and Nat (**k**) conditions (total *N* = 736). Time points for which the response to the figure was significantly greater than the response to ground are indicated by the horizontal line above each plot (p < 0.01, *t*-test).

The online version of this article includes the following figure supplement(s) for figure 5:

**Figure supplement 1.** Mean time courses of responses across the population to figure, ground, and border in areas V1, LM, and RL.

## Mouse segmentation is well modeled by a deep network

How could neural signals in the mouse support linear decoding of figure position in the Cross and Iso conditions despite lack of explicit figure-ground and border-ownership cells? To address this, we tested different neural encoding models for how well they could explain the observed decoding performance. We first simulated a population of 25,000 simple cells with varied receptive field size, location, orientation, preferred phase, and spatial frequency (see Methods). Each unit consisted of a linear filter followed by linear rectification and additive Gaussian noise (*Figure 7a*). We refer to this model as the 'feedforward LN model'. We attempted to decode figure position from this model using the same procedures as we used for analyzing the neural data (*Figure 6a*). Surprisingly, we found that we could robustly decode figure position for the Cross condition, though not for the Iso and Nat conditions (*Figure 7b*). It is widely assumed that figure-ground segregation (i.e., detecting the location of the square in the displays in *Video 2*) cannot be accomplished through purely local linear filters. How could a simple feedforward LN model decode figure position when the local stimulus at the figure center is identical to that of ground after pooling across conditions? We realized that to decode figure position, one need not rely on signals from the center of the figure; instead, one can use signals at the edges, and simple cells can readily localize orientation discontinuities such as those present in the Cross condition. This underscores an important point: the Cross stimulus completely fails as a behavioral marker for a nonlinear figure-ground segmentation process (see also *Vinken and Op de Beeck, 2021*).

We next modeled orientation-dependent surround modulation, a previously reported nonlinear interaction in mouse visual cortex (*Wang et al., 2011*; *Self et al., 2014*; *Keller et al., 2020a*). To simulate orientation-dependent surround modulation, we added a divisive term such that $response = response_{feedforward}/(1 + \beta * pearson(\vec{V}, \vec{V}_{out}))$, where $pearson(\vec{V}_{in}, \vec{V}_{out})$ is the correlation between the mean orientation energy within a cell's receptive field ($\vec{V}_{in}$), compared to that in the surround ($\vec{V}_{out}$). This surround model behaved similar to our feedforward LN model, failing to capture the texture dependence of the neural and behavioral data from the mouse (*Figure 7c*). This is not surprising, since the nonlinear interaction in this model depends on an orientation discontinuity, which was absent from the Iso condition. These results held generally across a range of noise levels (*Figure 7—figure supplement 1*).

Finally, we hypothesized that while orientation-dependent surrounds might be an insufficient nonlinearity to explain the mouse's behavioral and neural data, a deep convolutional network (DCN) trained on object recognition might develop many nonlinearities useful for performing the figure localization task. For example, common objects in cluttered scenes can resemble either the Cross or Iso conditions. We ran our stimuli through Vgg-16 pre-trained on ImageNet to perform object classification (*Figure 7d*) and analyzed responses in convolutional layers 1–5 (*Figure 7e*; *Simonyan and Zisserman, 2014*). We then tested decoding performance exactly as for the feedforward LN and surround models by randomly drawing subsamples of cells from a given layer. The performance of the

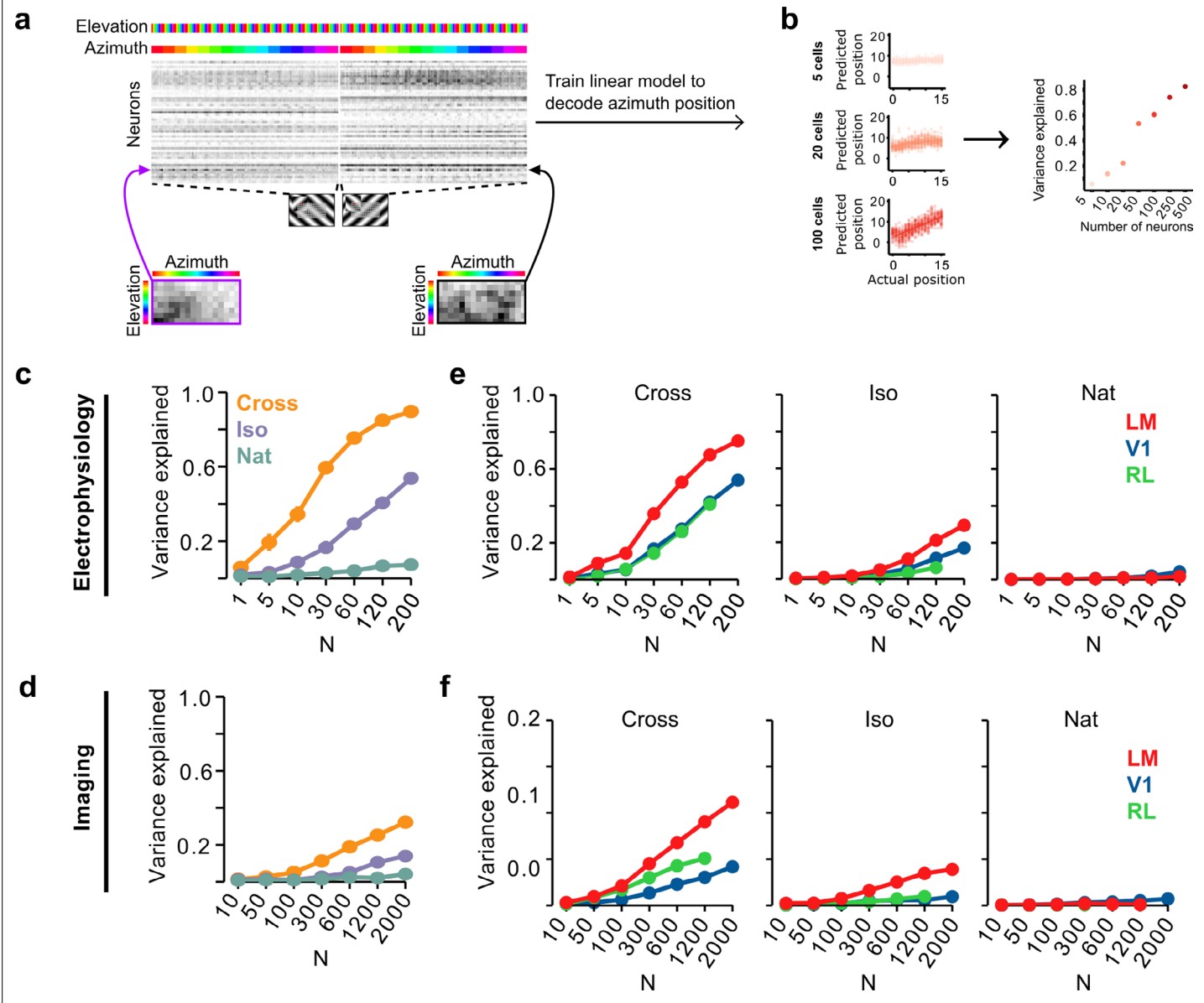

**Figure 6.** Decoding figure position from neural responses. (**a**) Schematic of approach for decoding figure position from neural population responses. For each neuron, figure response maps for both types of stimuli from a given texture condition (Cross, Iso, and Nat) were pooled, and reshaped into a 1-d vector, producing a population matrix of N neurons × 128 positions; the population response matrix for the Cross condition is shown. A linear decoder for figure azimuth position was then learned with cross-validation from the population response matrix using 50% of trials for training and the remaining 50% of trials for testing. (**b**) A linear readout was computed for a given random sample of *N* neurons, shown here for 5 (top), 20 (middle), and 100 (bottom) neurons. Each dot plots the actual azimuth bin (*x*-axis) against the predicted bin (*y*-axis). Mean explained variance was then computed across 50 repeated samples and plotted as a function of number of neurons (right). (**c**) Variance explained by decoded azimuth position as a function of number of neurons used to train the decoder for each of the different texture conditions (electrophysiology data). The most robust position decoding was obtained for Cross (orange), followed by Iso (violet) and then Nat (green). Error bars represent standard error of the mean (SEM) (total $n = 736$; V1: $n = 260$; LM: $n = 298$; RL: $n = 178$). (**d**) Same plot as (**c**) but for deconvolved calcium imaging data (total $n = 11,635$; V1: $n = 7490$; LM: $n = 2930$; RL: $n = 1215$). (**e**) Same data as in (**c**), but broken down by both texture condition and visual region. LM (red) consistently showed better positional decoding than either V1 (blue) or RL (green). Error bars represent SEM. (**f**) Same as (**e**) but for deconvolved calcium imaging data.

DCN matched the mouse's neural and behavioral data well: performance was best for Cross, followed by Iso, and then Nat, with this preference emerging in mid to late layers of the network. These results held generally across a variety of noise levels (*Figure 7—figure supplement 2*). Thus a DCN replicated the rank ordering of the mouse's behavior and neural decoding performance (Cross > Iso >

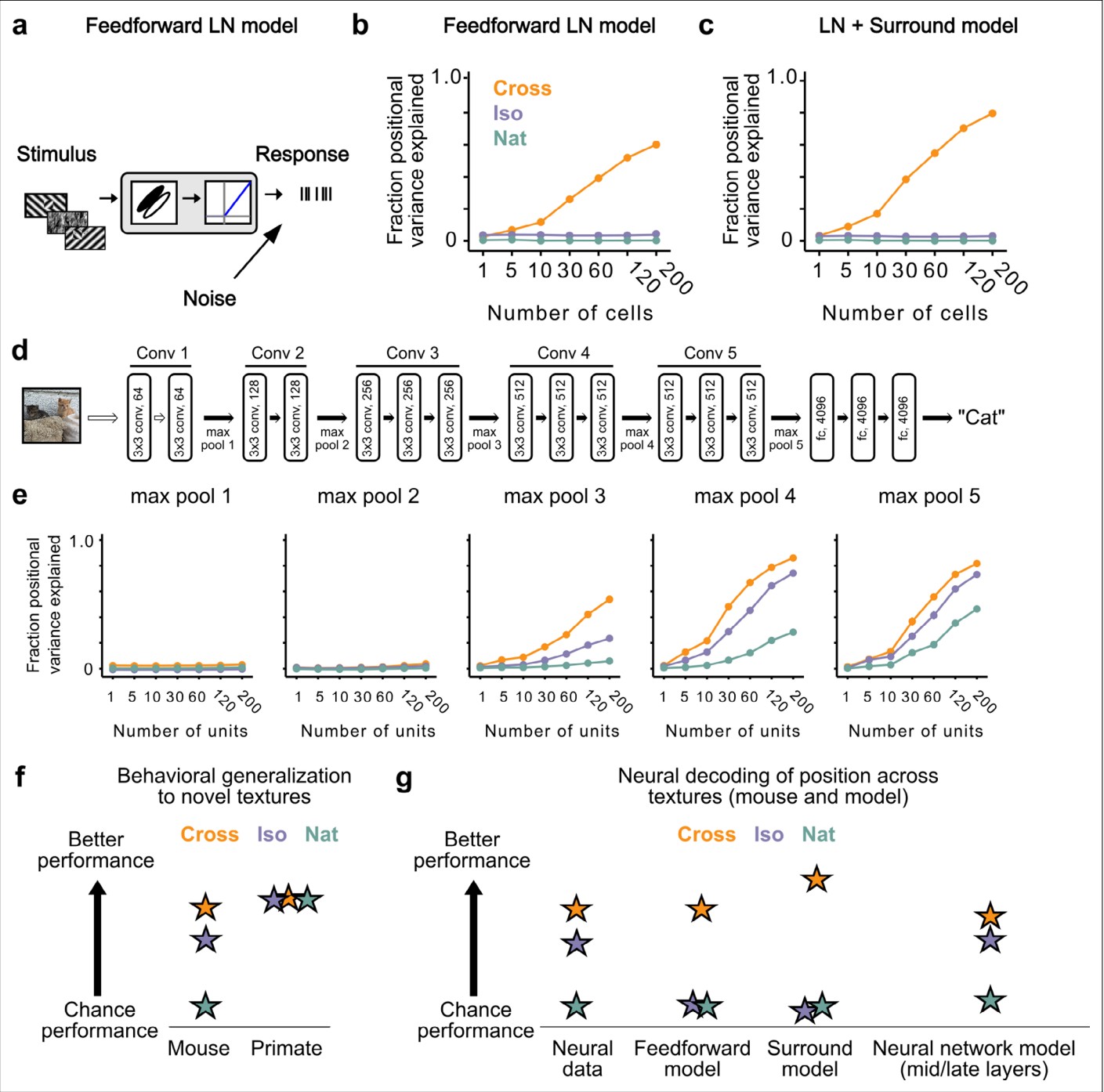

**Figure 7.** Mid to late layers of a deep network recapitulate mouse neural and behavioral performance on figure position decoding across texture conditions. (**a**) Schematic of the feedforward linear–nonlinear (LN) encoding model (see Methods). The stimulus was passed through a Gabor filter, followed by a rectifying nonlinearity, and then a Poisson spiking model, with noise added to responses to simulate population response variability (e.g., due to non-sensory signals such as movement or arousal). We ran the same stimuli (128 positions × 6 conditions) through the model that we used for electrophysiology and two-photon imaging (*Figure 4a*). (**b**) Positional decoding performance, quantified as variance explained by decoded azimuth position, as a function of number of neurons in the feedforward LN model. Cross (orange) positional decoding was robust, while both Iso (violet) and Nat (green) were extremely poor, in contrast to electrophysiology (*Figure 6c*) and imaging (*Figure 6d*) results. Noise variance was set to twice the network-level firing rate within a condition here and in (**c, e**) below; for a full sweep across noise parameters, see *Figure 7—figure supplement 1*. Error bars represent standard error of the mean (SEM). Small random offset added for visualization purposes. (**c**) Adding an orientation-dependent divisive term to the LN model to mimic iso-orientation surround suppression (*Figure 7—figure supplement 3*; LN + surround model) yielded more robust decoding in the Cross condition (orange), but did not improve decoding in the Iso (violet) or Nat (green) conditions. For a full sweep across noise

*Figure 7 continued on next page*

*Figure 7 continued*

parameters, see *Figure 7—figure supplement 1*. Error bars represent SEM. Small random offset added for visualization purposes. (**d**) Architecture of a pre-trained deep neural network (VGG16) trained on image recognition (*Simonyan and Zisserman, 2014*). Five convolution layers are followed by three fully connected layers. (**e**) Positional decoding performance increases throughout the network with most robust decoding in layer 4. In mid to late layers (3–5) of the deep network, decoding performance was best for Cross (orange), followed by Iso (violet) and then Nat (green), mirroring mouse behavioral performance (*Figure 2c, i1*) and neural data (*Figure 6c, d*). For a full sweep across noise parameters, see *Figure 7—figure supplement 2*. Error bars represent SEM. (**f**) Schematic summary of behavioral results. Mice showed texture-dependent performance in a figure localization task, with Cross > Iso > Nat, despite the presence of a common motion cue for segmentation in all conditions. In contrast, primates showed no dependence on carrier texture and were able to use the differential motion cue to perform the task. (**g**) Schematic summary of neural and modeling results, Positional decoding from neural populations in V1, LM, and RL mirror the textural dependence of the behavior, with Cross > Iso > Nat. This ordering in performance was not captured by a feedforward LN model or an LN model with surround interactions. However, it emerged naturally from nonlinear interactions in mid to late layers of a deep neural network.

The online version of this article includes the following figure supplement(s) for figure 7:

**Figure supplement 1.** The effect of noise on position decoding for feedforward LN and surround models.

**Figure supplement 2.** The effect of noise on position decoding for intermediate layers of VGG16.

**Figure supplement 3.** Modeling orientation-dependent surround interactions.

Nat). This suggests the possibility that the mouse visual system may use similar nonlinear interactions as in a feedforward deep network to accomplish object detection.

## Discussion

We have shown that mice and primates segment visual input using fundamentally different strategies. Unlike primates, mice are unable to detect figures defined purely by opponent motion, and hence mouse segmentation behavior is strongly dependent on texture cues (*Figure 7f*). Indeed, when mice were forced to detect figures defined purely by opponent motion for a limited number of patterns, they adopted a strategy of brute force pattern memorization (*Figure 2e–g*). The strong texture dependence of mouse object detection behavior was consistent with neural signals recorded in mouse visual areas V1, RL, and LM, and could be explained by a simple feedforward deep network model lacking explicit segmentation capabilities (*Figure 7g*).

When we tested three additional species, the macaque, mouse lemur, and treeshrew, using the same paradigm, we found that only the two primate species could perform segmentation using opponent motion. It was especially surprising that the mouse lemur, a tiny (~60–80 g) prosimian primate species (*Ho et al., 2021*), could segment purely motion-defined figures well above chance, while the treeshrew (~120–200 g), an animal with a much more highly developed visual system than the mouse (*Wong and Kaas, 2009*; *Mustafar et al., 2018*; *Van Hooser et al., 2013*), could not. We emphasize again that the differences cannot be attributed to differences in visual acuity between the species for multiple reasons, including that mice had no difficulty detecting an otherwise identical moving square on a static background, and treeshrews and mouse lemurs have highly similar visual acuities (*Ho et al., 2021*). Overall, our findings reveal a fundamental difference in the computational strategy used by mice/treeshrews versus primates for visual segmentation and suggest that surface perception from accretion–deletion may be a capability unique to primates. We believe this is highly significant because visual surface representation is a fundamental step in primate visual processing (*Nakayama et al., 1995*; *Roe et al., 2012*; *Tsao and Tsao, 2022*), and accretion–deletion (*Figure 1b, c*) has been recognized since the seminal work of J.J. Gibson as *the most powerful cue* supporting visual surface representation (*Gibson, 1979*). In particular, among all cues to surface organization, accretion–deletion is unique in its (1) high reliability, occurring only at true object borders and never at internal texture borders, (2) robustness to object texture, and (3) computational power, supporting not only segmentation but also invariant tracking without requiring any learning or prior experience (*Tsao and Tsao, 2022*).

We were inspired by previous rodent behavioral studies that have sought to carefully characterize the visual capabilities of mice and rats, testing behaviors such as transformation-tolerant object recognition and natural scene discrimination (*De Keyser et al., 2015*; *Vermaercke and Op de Beeck, 2012*; *Yu et al., 2018*; *Zoccolan, 2015*; *Zoccolan et al., 2009*; *Schnell et al., 2019*). In particular, consistent with our finding that mice cannot detect naturalistic figures defined by opponent motion,

Keyser et al. found that rats could not learn to detect a bar defined by a grid of Gabor patches moving counterphase to ones in the background (*De Keyser et al., 2015*) (note, however, their stimulus did not contain accretion–deletion, the cue we were especially interested in for reasons explained above). Overall, our results add to a growing body of work showing that rodent and primate vision differ in essential ways beyond visual resolution.

Our finding that figure-ground signals exist in mouse visual cortex (*Figures 3e, f, 4*, and *5d, i*), but are strongly dependent on texture (*Figures 4 and 5a, b, e, i–k*), is consistent with previous studies (*Schnabel et al., 2018a*; *Kirchberger et al., 2020*; *Schnabel et al., 2018b*; *Keller et al., 2020b*). Optogenetic perturbation studies have further demonstrated that these signals are behaviorally relevant for figure detection (*Kirchberger et al., 2020*) and require feedback (*Kirchberger et al., 2020*; *Keller et al., 2020a*). Thus overall, it seems clear that mouse visual cortex shows orientation-dependent surround modulation (*Self et al., 2014*; *Keller et al., 2020a*) which can support texture-based figure-ground segmentation behavior. However, importantly, our results show that mice lack a general, texture-invariant mechanism for surface segmentation, unlike primates. One caveat is that we did not directly record from corresponding primate visual areas using the exact same stimuli as we used for the mice in this study; however, the stimuli we used were highly similar to those reported in the macaque literature (e.g., *Lamme, 1995*).

The inability of mice to detect figures defined purely by opponent motion was rather surprising, as cells selective for local motion (distinct from global retinal image drift due to fixational eye movements) have been reported in the retinas of both rabbits and mice (*Kim et al., 2015*; *Olveczky et al., 2003*). How could a signal present in the retina not be used by the animal for segmentation? First, in our experiments, figure and ground moved exactly in counterphase, a condition that the retinal object-motion cells are unable to detect (*Olveczky et al., 2003*). Furthermore, retinal studies have generally used sinusoidal gratings, and it remains unclear how responses of retinal object-motion cells might generalize to arbitrary textures such as those used in our Nat task. Finally, we note that object localization in the Nat task required perception of *differential* motion cues. We confirmed that mice were readily able to detect the same moving figures against stationary backgrounds (*Figure 2—figure supplement 2c*). It is possible that retinal object-motion cells are adapted for this latter condition—whose handling may be sufficient to ensure mouse survival—while different evolutionary pressures led to emergence of a more sophisticated segmentation mechanism in primates.

The distinction between mouse and primate segmentation behavior has an intriguing parallel in the difference between deep network and human object classification behavior. In recent years, deep network models of vision have started to achieve state of the art performance on object classification tasks. However, the lack of an explicit segmentation process in these networks leads to susceptibility to adversarial examples in which noise patterns that lack any surface structure are classified as objects with high confidence (*Goodfellow et al., 2014*; *Szegedy et al., 2013*). Furthermore, recent work has shown that deep networks, unlike humans, fail to exploit global image features such as object boundaries and shape to perform classification, instead relying much more strongly on texture features (*Brendel and Bethge, 2019*; *Geirhos et al., 2018*). Thus it is clear that a major difference between deep networks and the primate visual system lies in their segmentation capabilities. Our finding of strong texture dependence in mouse segmentation behavior suggests that mice may adopt a visual strategy more similar to deep networks than primates do (*Figure 7g*; see also *Vinken and Op de Beeck, 2020*), though further detailed circuit dissection will be necessary to test this conjecture.

One may wonder whether the difference between mouse/treeshrew versus primate vision that we have uncovered is truly of ethological significance, since all of the species studied were able to segment objects from the background using texture or texture-invariant motion cues on a static background, and in the natural world, backgrounds are usually immobile. We underscore that we are not claiming that accretion–deletion is the only cue to segmentation. Rather, its chief importance is that it can enable texture-invariant segmentation and tracking *without learning*, thus it can provide a rich stream of training examples to self-supervise learning of other cues (*Tsao and Tsao, 2022*; *Chen et al., 2022*). In this respect, the discovery that non-primate brains are incapable of using accretion–deletion is highly ethologically significant, because it means a key *computational mechanism* for self-supervised learning of a physical world model may be unique to primates.

Objects are the fundamental building blocks of a primate's model of the world. An object is first and foremost a *moveable* chunk of matter, that is, a segmentable surface. The remarkable trajectory

of the human species has been variously attributed to language, tool use, upright posture, a lowered larynx, opposable thumbs, and other traits (*Leakey, 1996*). We close with a speculation: it may not be entirely implausible that possession of machinery for cue-invariant visual object segmentation played a role in setting the human trajectory. By enabling self-supervised learning of a rich and accurate physical model of the world inside the primate brain, this perceptual machinery, seemingly less developed in all non-primate species tested so far including both mice and treeshrews, may have supplied the foundation for subsequent human capabilities requiring a hyper-accurate model of objects in the world—including tool use, causal understanding, and general intelligence.

## Methods

### Animal statement

The following animals were used in this study: adult mice 2–12 months old, both male and female; adult treeshrews 7–18 months old, both male and female; adult mouse lemurs 2–3.5 years, both male and female; and adult macaques 3 and 7 years old, male. All procedures on mice, macaques, and treeshrews were conducted in accordance with the ethical guidelines of the National Institutes of Health and were approved by the Institutional Animal Care and Use Committee at the California Institute of Technology.

Mouse lemur experiments were in accordance with European animal welfare regulations and were reviewed by the local ethics committee (Comite d'éthique en expérimentation animale No. 68) in Brunoy, France, by the ethics committee of the University of Geneva, Switzerland and authorized by the French 'Ministère de l'education nationale de l'enseignement supérieur et de la recherche'.

### Transgenic animals

For imaging experiments, we used a cross between a CamKII::ttA mouse (JAX: 00310) with a tetO:G-CaMP6s (JAX: 024742) to target expression to cortical excitatory neurons. For electrophysiology experiments, we used a Thy1::GCamp6s 4.12 (JAX: 025776). Behavioral experiments were carried out with a combination of Thy1 and C57BL/6 animals. We back crossed all lines to C57BL/6.

### Surgical procedures

The cranial window and headplate procedures were based on *Wekselblatt et al., 2016* with some modifications as described below.

### Headplate surgery

For both electrophysiology and imaging experiments, a stainless steel headplate was attached to the animal's skull in a short procedure. Animals were anesthetized using isoflurane (3% induction; 1.5–2% maintenance) in 100% $O_2$ (0.8–1.0 l/min) and positioned in a stereotax using earbars placed just below the ear canal for stability. The animals were given subcutaneous injections of the analgesic Ketoprofen (5 mg/kg) and 0.2 ml saline to prevent postoperative dehydration. Body temperature was maintained at 37.5°C by a feedback-controlled heating pad; temperature and breathing were monitored throughout surgery. Sterilized instruments and aseptic technique were used throughout. Sterile ocular lubricant (Puralube) was applied at the beginning of each surgical procedure. Scalp hair was removed using an electric shaver, and the surgical site was cleaned using a combination of dermachlor and chlorohexidine. A scalp incision was made using #3 scissors (FST) and the periosteum was pulled back using forceps. The back neck muscles were retracted to make room for a either an 8 or 10 mm circular opening headplate which was affixed to the skull using either metabond (Parker) or dental acrylic (OrthoJet). A combination of vet bond and cyanoacrylate-based glue was applied to the skull to both protect the skull surface from infection and to provide a clear surface through which to perform widefield imaging to identify cortical visual areas in electrophysiology experiments.

### Craniotomy surgery/window implantation

After allowing the animal to recover at least 1 week from the headplate surgery a craniotomy procedure was performed to either allow for acute implantation of an electrode or to install a glass coverslip for chronic imaging.

Animals were anesthetized using isoflurane (3% induction; 1.5–2% maintenance) in 100% $O_2$ (0.8–1.0 l/min) and positioned in the stereotaxic frame affixed by the headplate attached previously. The animals were given subcutaneous injections of the analgesic Ketoprofen (5 mg/kg) and 0.2 ml saline to prevent postoperative dehydration. Body temperature was maintained at 37.5°C by a feedback-controlled heating pad; temperature and breathing were monitored throughout surgery. Sterilized instruments and aseptic technique were used throughout. Sterile ocular lubricant (Puralube) was applied at the beginning of each surgical procedure.

For imaging experiments, a 4–5 mm craniotomy was cut out centered at +0.5 mm from lambda and +2.75 mm from midline on the right hemisphere. Care was taken to minimize bleeding, and any bleeds in the skull during drilling were covered with wet gelfoam (Pfizer) until they resolved. After careful removal of the bone flap, a durotomy was performed and the exposed brain was covered in a 1:1 mix of artificial dura (Dow Corning 3-4680). A sterile 4–5 mm coverslip was then pressed into the opening and sealed in place using a combination of cyanoacrylate-based glues. The remaining parts of exposed skull in the headplate well were then covered with black dental acrylic for light blocking purposes and to prevent infection.

For electrophysiology experiments, retinotopic mapping was performed prior to performing any craniotomy, resulting a vasculature and field sign map to identify vasculature landmarks corresponding to either V1, LM, or RL. Once such landmarks had been identified, a small (<1 mm) craniotomy was performed on the morning of each experiment. The craniotomy was sealed with KwikSil (WPI) and animals were allowed to recover for at least 3 hr before subsequent recording experiments.

## Visual stimuli

Visual stimuli for electrophysiology/imaging were presented on a 32-inch monitor (Samsung 32 inch lcd screen; 40 × 71 cm); linearized by eye to correct for gamma (mean luminance 50 cd/m²), oriented tangentially 18 cm from the mouse's left eye (figure size: ~30°) for all experiments except those shown in *Figure 3f* for quantifying reproducibility which were carried out at 27 cm from the eye (*Figure 3b*).

For all electrophysiology and imaging experiments, attempts were made to re-center the monitor on the general location of the receptive fields across the population to maximize the number of distinct figure and ground trials in a given experiment.

## Figure-ground/border-ownership stimulus

The stimuli used to characterize FGM and BOM of each cell consisted of a sinusoidal grating of 0.06 cpd, oriented at 45° or 135°. The figure was 27° in size. The horizontal positions surveyed spanned 45° (~3° shift per position), while the total elevation positions surveyed varied across 23° (~3° shift per position). For the Iso and Nat conditions where the texture defining the figure was identical within a given condition, the figure was generated at each position using the same texture that previously defined the background, thus creating a 'pop out' effect for the stimulus (see *Video 2*). Both figure and background moved out of phase with a sine wave of amplitude 14 pixels (3.5°) and temporal frequency of 1 cycle/s.

For the naturalistic texture used in electrophysiology/imaging experiments, we used synthetically generated naturalistic textures identical to those used in *Freeman et al., 2013*. We did not regenerate the textures between experiments and instead chose two random textures which we presented for the two Nat conditions, similar to how two orthogonal orientations were presented for Cross and Iso conditions. These same textures were also used for Nat condition in the behavioral experiments in *Figure 2*, with the exception of the new textures in *Figure 2f*, which were taken from the naturalistic textures database. The naturalistic textures presented in the behavior in *Figure 2—figure supplement 2* were also from the naturalistic textures database. For both electrophysiology and two-photon imaging experiments, the 6 conditions (Cross, Iso, Nat × 2 conditions) were pseudo-randomly interleaved, i.e., Cross/Iso/Nat were presented in a random order, and the two within-condition repetitions (e.g., two orientations or textures) were presented consecutively but also randomized.

## RF mapping

Receptive fields were mapped for neurons under two-photon using an isolated drifting Gabor patch stimulus: a patch of ~6° containing a drifting Gabor appeared in one of three different orientations (45°, 180°, and 315°) and two directions at a random position. We repeated this procedure for 9

repeats and at 16 by 9 positions and then collapsed responses across all trial types to compute a spatial PSTH. We then fit a 2D Gaussian to the response and classified neurons as having a significant receptive field fit if the goodness of fit exceeded the goodness of fit for at least 99 out of 100 bootstrapped trials where spatial location was shuffled across trials.

For electrophysiology, we used a sparse noise stimulus to map the spatial RFs in all three areas. We divided the screen into a 18 × 32 grid of squares, each 3.5° on a side (each grid square was 2.2 × 2.2 cm, 18 cm from the mouse's left eye). In each stimulus frame, one grid square, black or white, was presented alternately. The stimulus contained 8000 frames in total and was presented at 10 Hz. RFs were computed by spike triggered average (STA). The analysis window for STA was 30–100 ms. Then we used 2D Gaussian to fit the RFs. *The goodness of fit* (GOF) = 1 − MSE/ *Var(RF)*, in which *MSE* is mean square error of the fitting = $\frac{1}{n}\sum_{i=1}^{n}\left(RF\left(i\right)-G\left(i\right)\right)^2$ , *RF* is the receptive field of the cell, *G* is the 2D Gaussian fit of the RF, *Var(RF)* is the variance of the *RF*, $Var\left(RF\right)=\frac{1}{n}\sum_{i=1}^{n}\left(RF\left(i\right)-mean\left(RF\right)\right)^2$. The cells with RF GOF above 0.1 were include in subsequent analyses.

We limited our analysis for FGM and BOM in *Figure 5* to neurons that had significant receptive field fits and used the central position of that fit as the neuron's inferred RF center. For population neural decoding analyses, we used all neurons regardless of receptive field fits.

## Mouse behavior

We trained 16 adult mice aged 8–12 weeks (4 × C57bl/6; 12 × Thy1-Gcamp6s mice) co-housed 4 to a cage with access to a running wheel. Mice were housed under reversed light–dark cycle (dark cycle from 10 am to 9 pm), and training was performed during the dark cycle. Mice were water restricted before, during, and after training days. The weight of the animals was continuously monitored to make sure no mouse dropped below 85% of their pre-water-restriction weight.

Mice were trained for 6 days/week, with a single session per day. Each mouse performed around 200–450 trials per day depending on the behavioral task. All training was carried out a commercially available touchscreen operant chamber in a sound-attenuating box (Bussey-Saksida Touch Screen Operant Chambers/Lafayette Instrument).

Visual stimuli were presented on a touchscreen monitor (24 × 18.5 cm, *W × H*, 800 × 600 pixels) (Lafayette Instrument). In all experiments, size of figures is given in pixels as the animals were freely moving and thus no estimate can be made of the size of the figure in visual degrees during the behavior as it will vary depending on the animal's position. We generated stimulus movies of size 100 pixels high × 200 pixels wide, with square figures centered at either 50 or 150 pixels and square side of 50 pixels. Both figure and background moved out of phase with a sine wave of amplitude 14 pixels and temporal frequency of 1 cycle/s.

One side of each chamber was a touch screen monitor, and on the other side was a reward tray with a water spout. Mice had to touch the left or right side of the screen based on the location of the figure. The water delivery was cued by a white light illuminating the spout. An IR sensor detected when the mouse collected the water reward and determined the start of the next trial. For incorrect responses, a white noise audio stimulus was played for 3 s, followed by a ten second timeout. After the time-out period, the stimulus of the identical trial was presented again until the mouse made the correct touch. This shaping procedure was used to prevent biases in the behavior. All trials following the first incorrect response were not included in the analysis, as other similar studies have done (*Schnabel et al., 2018a*; *Stirman et al., 2016*).

We first trained the animal to become familiar with the touch screen and the task rules through a luminance shaping procedure. During this procedure, the mice learned to discriminate a white square moving on a black background and touch the side of the screen that displayed the square. In this task, they also learned that a correct touch was associated with a water reward. The mice started their subsequent training steps after reaching 80% on this task.

### Standard 2-pattern discrimination

For these experiments, there were 4 possible stimuli (2 orientations/natural noise × 2 positions) (*Figure 2b*, *Figure 2i1*, *Figure 2—figure supplement 2b1*). Gratings were presented at two orientations, 45° and 135° (0.25 cycles/cm).

### 5-Pattern discrimination

Animals were tested on this after training on the 2-pattern task (*Figure 2i1*). The stimuli were the same as for the 2-pattern discrimination, except we introduced five novel orientations or naturalistic noise patterns. For gratings, the five novel orientations were 22.5°, 67.5°, 90°, 112.5°, and 157.5°. For the Nat condition, we used five new natural noise patterns that shared the same Fourier statistics.

### 30 natural textures

After finishing training on 2- and 5-pattern discrimination tasks, mice were trained on 30 natural textures in Iso and Cross configurations (*Figure 2—figure supplement 1*). We first tested them on seven textures (set A) to measure the baseline performance. We then trained them on 30 Iso and Cross natural textures (set B). After training, they were tested on seven textures again (set A). For experiments involving natural textures we randomly selected textures from the describable textures dataset (DTD; https://www.robots.ox.ac.uk/~vgg/data/dtd/). We converted all textures to grayscale.

### Grating to noise shaping

We generated a gradual morph stimulus that changed oriented gratings into naturalistic noise, with the goal being to train the animals to use the motion cue (*Figure 2e*). In total, there were 10 stages in this shaping procedure. At each stage, the stimulus represented a weighted sum of the grating and noise, and the weight was changed 10% from the previous stage for both grating (decrease) and noise (increase). For example, in stage 3, the weight assigned to grating was 70% and the weight for noise is 30%. By stage 10, the weight for grating was 0% and for noise was 100%. At each stage, figure texture was flipped vertically and presented at a random height (top, middle, and down) to discourage use of local cues in the task.

### Static version of tasks

We tested mice on static versions of all conditions where only a single frame of the stimulus was presented (*Figure 2g, h*). The static stimulus was the frame that had the largest phase difference between figure and background.

### New texture task

After mice learned the grating to noise task, they were tested on seven unseen new natural textures (*Figure 2f*).

### Background static natural textures

We trained the mice to detect a figure moving horizontally on a static background. This stimulus was identical to the Nat condition, the only difference was that the background was static (*Figure 2—figure supplement 2c*).

## Treeshrew behavior

Four adult treeshrews (three male, one female; age: 7–18 months old) bred and raised at Caltech were trained to perform the figure-ground segregation task. Animals were singly housed in a 12 hr:12 hr light:dark cycle. They were not food or water restricted, but free access to water was limited during the 4 hr prior to training each day. Training was performed during the light cycle in a custom-made behavioral arena (30 × 30 × 25 cm) containing three optical lickports (Sanworks) situated in a custom-built behavior box. Drops of 100% apple juice rewards were provided upon poking at the appropriate lickports and in some cases for trial initiation. Images were presented on a Sunfounder 10.1″ Raspberry Pi 4 (1280 × 800) screen and controlled using Bpod hardware and Python software. After an initial shaping step in which animals learned to use the lickports in a luminance detection task (2–3 days), training for Cross, Iso, and Nat conditions was performed for 4 consecutive days (five orientations or naturalistic textures), with generalization test sessions on the fifth day (two different orientations or naturalistic textures) (*Figure 2—figure supplement 2a, b2*). We reversed the number of train/test patterns compared to what was used for the mice (*Figure 2i1*) because we reasoned that animals might be more likely to generalize if given more patterns for training. We had performed the mouse experiments initially, noticed the memorization approach, and were trying to avoid this behavior in

treeshrews. This also means that the naturalistic train condition presented to treeshrews was harder than that for mice (five orientations for treeshrews vs. two orientations for mice in the training set).

## Mouse lemur behavior

Four adult mouse lemurs (three male, one female; age: 2–3.5 years) bred and raised in the 'Mouse Lemur Platform' (authorization number E-91-114-1) of the 'Museum National d'Histoire Naturelle' in Brunoy, France (UMR MECADEV CNRS/MNHN 7179) were trained to perform the figure-ground segregation task. Animals were co-housed 2–3 per cage in a reversed long-day (14:10 light:dark) cycle. They were food restricted, with their body weight maintained above 60 g, but had free access to water. Training was performed during the dark cycle in a custom-made behavioral arena (20 × 20 × 30 cm) containing three optical lickports (Sanworks) situated in a sound-attenuating box. Drops of liquid food rewards (standard food mixture composed of banana, cereal, milk, and egg) was provided upon poking at the appropriate lickports. Images were presented on a Dell P2414H (1920 × 1080, 60 Hz) screen and controlled using Psychopy and Matlab software. Training and testing followed the same paradigm as for treeshrews (*Figure 2—figure supplement 2a, b3*).

## Macaque behavior

Two head-fixed rhesus macaque monkeys were trained to indicate whether a square was on the left or right side of a screen. Movie stimuli were shown on an LCD screen in pseudo-random succession for 2 s ON time each, without any OFF period. The stimuli were shown across the full screen (23° in height and 37° width) and contained a square of 9° length on either the left or right side. Monkeys received a juice reward for fixating within the 9° square region for at least 1 s. Eye position was monitored using an infrared eye tracking system (ISCAN). Each monkey performed only one or two sessions of 1 or 2 hr each. In the beginning of the first session, behavior was shaped by training the monkeys on the luminance squares only until they reached 90% correct performance. Prior to this, the monkeys had been trained only to fixate. Both monkeys learned this within the first session and subsequently performed the task with all other stimuli presented in pseudo-random succession (*Figure 2—figure supplement 2b4*). Stimuli included static and moving luminance squares, cross-orientation and iso-orientation gratings, and natural textures.

For offline analysis, we computed the percentage of correct trials where monkeys were fixating the correct square location for at least 1 s, including only trials where monkeys were looking at the screen and not closing the eyes.

## Wide-field imaging

Prior to all electrophysiological and imaging experiments, a reference vasculature image and field-sign map was acquired under a custom-built widefield epi-fluorescence microscope. The microscope consisted of two Nikon 50 mm f1.4 lens placed front to front with a dichroic, excitation, and emission filters (Semrock) in between. Light was delivered via a blue LED light source (Luxeon Star) and images were acquired with a CMOS camera (Basler). Images were acquired at 10 Hz and were triggered on every third frame of a 30 Hz retinotopic mapping stimulus (drifting bar; trial period of 0.1 Hz) to ensure proper timing between stimulus and acquisition. Retinotopic mapping stimulus consisted of a drifting 10° bar of binarized 1/f noise (*Wekselblatt et al., 2016*), which cycled with a period of 0.1 Hz. Elevation and azimuth maps were computed using a Fourier decomposition of the stimulus and plotting preferred phase at the stimulus frequency (*Kalatsky and Stryker, 2003*).

## Two-photon imaging

We began imaging sessions ~2 weeks after surgery. We used a resonant, two-photon microscope (Neurolabware, Los Angeles, CA) controlled by Scanbox acquisition software (Scanbox, Los Angeles, CA). Imaging was through a ×16 water immersion lens (Nikon, 0.8 NA) at an acquisition rate of 15.6 Hz at depths ranging from 150 to 250 μM from the surface corresponding to layer 2/3. Mice were allowed to run freely on a spherical treadmill (styrofoam ball floated with air).

We ran a minimum of 7 stimulus conditions in all sessions (RF mapping +6 conditions) with a short (<3 min) break between each imaging session. Most sessions lasted less than 75 min. For analysis, all of the movies from each session were aligned to a common mean image using a non-rigid registration pipeline (Suite2P). Briefly, all movies were aligned to a common reference frame

by estimating a semi-rigid transformation of all frames. The mean activation image was generally used at the reference frame. After alignment, an approach to jointly estimate filters and traces corresponding to a generative model as outlined in the original suite2P paper was applied (*Pachitariu et al., 2017*). These corresponding fluorescence traces are likely heavily contaminated with neuropil from adjoining processes and nearby pixels. Due to the retinotopic organization of the visual cortex, this sort of spatial correlation is likely to pose significant problems in trying to address minute differences in spatial coding of neurons in cortical space and visual space. To help further separate out these signals we thus went one step further and attempted deconvolution of these signals into a more nonlinear estimate of putative 'spiking' activity which can help disentangle the neighboring contributions.

Baseline correction was carried out via subtraction with a sliding window of baseline estimation corresponding to the 8th percentile of the raw fluorescence within a 60-s window of activity. This baseline-corrected trace was then further passed into the deconvolution algorithm in suite2P, resulting in an instantaneous and nonlinear estimate of activity that was used for all subsequent analysis.

## Matching cells across days

Cells were tracked across days by first re-targeting to the same plane by eye such that the mean fluorescence image on a given day was matched to that on the previous day, with online visual feedback provided by a custom software plugin for Scanbox. Then, a registration correction was computed from 1 day to the other such that the two planes were aligned. After extracting cell identities and cell filter maps for both days independently, a Jaccard index $J(A,B)=(|A \cap B|)/(|AB|)$ was computed for a given cell filter on day 1 with all extracted filters on day 2. A Jaccard index >0.5 was considered the same cell on the next day. Note that the alignment between cells for 'matched' versus 'un-matched' was determined purely based on the morphology of the extracted filter map, after both days had been aligned to a common reference mean frame. The distributions of correlations between the spatial figure responses presented in *Figure 3g* is completely independent validation of the fact that these are the same cells, as nothing about the cell's activity was considered in aligning cells across days. This result points to the consistency of the spatial responses in the visual cortex as a substrate for inferring figure position.

## Electrophysiology

Electrophysiology experiments were carried out with an acutely inserted 64 channel silicon probe from Sotiris Masmanidis (*Yang et al., 2020*) attached to a 4-axis manipulator (Siskiyou), that was amplified through a 128 channel headstage (Intan). Signals were sampled at 30 kHz and filtered through a bandpass filter (300–6000 Hz), then digitized by an open-ephys acquisition box (Open-ephys) and aligned to stimulus frames through the use of a photodiode. We used Kilosort (*Pachitariu et al., 2016*) for spike sorting of the data. The output of the automatic template-matching algorithm from Kilosort was visualized on Phy and then curated manually. Mixed single units and multi-units were included. We used two criteria to select for the cells included in our population analysis: (1) RF GOF >0.1; (2) Total spikes >100 for each stimuli. The probe was lowered into the brain at 5 µM/s and allowed to settle for a minimum of 5 min before experimental stimuli were presented. Animals were head-fixed for a maximum of 2 hr in any given experiment. Stimuli were presented as outlined in the Visual Stimuli section of the Methods.

## Analysis

All analyses were performed using custom scripts written in MATLAB (Mathworks) or Python using NumPy, SciPy, Pandas, seaborn, sklearn, and Matplotlib (*Hunter, 2007*; *McKinney, 2010*; *Pedregosa et al., 2011*; *van der Walt et al., 2011*; *Michael et al., 2018*).

### Trial-based response

For all trial-based analysis, we quantified the response for a given trial as the mean spike count 50–250 ms post trial onset. For imaging experiments, we use the deconvolved calcium trace where the response value was set to the mean across all frames of the trial.

## Positional decoding using linear regression

To quantify the amount of information present about figure position, we decoded the azimuth bin (positions 1–16) of each trial from a population of neurons using a ridge regression model and 50/50 cross-validation; results averaged across 100 iterations are reported for all data and modeling. To quantify the extent to which a single linear model could account for position across both orientations or textures, we pooled trial types within a stimulus condition (both orientations/textures for Cross/Iso/Nat).

Beta values (penalty term) for the ridge regression were computed with 50/50 cross-validation approach using the RidgeCV function from sklearn. As the beta values will be dependent on the number of regressors (neurons) in the model, this entire procedure was repeated for varying numbers of neurons to compute the decoding performance as a function of number of neurons in the decoder. All values reported for decoding correspond to the mean variance explained by the model over 100 iterations of the above procedure, and error bars correspond to 95% confidence intervals.

## Computing FGM and BOM

We limited analysis of FGM and BOM to electrophysiologically recorded neurons that satisfied two criteria: (1) they showed a statistically significant receptive field fit, and (2) the receptive field center was limited to a central portion of the screen (central 15° azimuth and 10° elevation). This second receptive field position criterion was to ensure a reasonable number of figure and ground or left and right trials with which to compute the FGM or BOM indices.

For each of the three conditions, we defined an FGM index as $FGM = \frac{(R_{Fig} - R_{Back})}{(R_{Fig} + R_{Back})}$, where $R_{Fig}$ is the mean response across the two patterns for the condition within the figure zone, defined as the $2 \times 2$ ($10° \times 10°$) grid of locations centered on the cell's receptive field (*Figure 5c*) and $R_{Back}$ is the mean response across the two patterns for the condition in the background zone, defined as all grid locations with distance greater than 1.5 * the receptive field width from the receptive field center. We computed bootstrapped p values using 500 shuffles where trial identity was randomized to establish a null distribution.

We quantified border-ownership selectivity in a similar way. We defined a BOM index as $BOM = \frac{(R_{Left\ border} - R_{Right\ border})}{(R_{Left\ border} + R_{Right\ border})}$, where $R_{Left\ border}$ is the mean response across the two patterns for the condition within the left border zone, and $R_{Right\ border}$ is the mean response across the two patterns for the condition within the right border zone (*Figure 5f*). We computed the significance of the modulation using a bootstrapped distribution as above.

## **Modeling**

### Feedforward model

We modeled neuronal responses using an LN model of simple cells, with the linear filter modeled by a Gabor function and linear rectification. We simulated responses to 100 different cell types using a classic simple cell model as a linear combination of receptive field with stimulus passed through a nonlinearity.

$$response_{feedforward} = f\left(g\left(x, y, \theta, \lambda, \gamma, \sigma, \omega\right) * stimulus\right)$$

where $g\left(x, y, \theta, \lambda, \gamma, \sigma, \omega\right)$ is a Gabor function:

$$g\left(x, y, \theta, \lambda, \gamma, \sigma, \omega\right) = e^{\frac{-x'^2 + \gamma^2 y'^2}{2\sigma^2}} cos\left(2\pi \frac{x'}{\lambda} + \omega\right), \begin{pmatrix} x' \\ y' \end{pmatrix} = \begin{pmatrix} cos\theta & sin\theta \\ -sin\theta & cos\theta \end{pmatrix} \begin{pmatrix} x \\ y \end{pmatrix}$$

with parameters sampled as follows: $\theta$ = rand(0, $\pi$), $\sigma$ = rand(2°, 7°), $1/\lambda$ = rand(0.05 cpd, 0.3cpd), $\omega$ = rand(0, $\pi$), $\gamma$ = 1. $f(x)$ represents a linear rectification (max(0,x)) to ensure positive rates and * represents the convolution operator.

### Surround model

We added a divisive term to the neural response computed from our feedforward LN model (*Figure 7—figure supplement 3*). This divisive term was not recurrent and instead can be most readily

interpreted as a center-surround interaction that would arise in the feedforward inputs from thalamus to cortex.

$$response = \frac{resopnse_{feedforward}}{1 + \beta * \rho(\vec{V}_{in}, \vec{V}_{out})}$$

where $\rho(\vec{V}_{in}, \vec{V}_{out})$ represents the Pearson correlation between the mean response of all neuron types within the receptive field ($<2\sigma$; $\vec{V}_{in}$) and all neuron types outside the classical receptive field ($>2\sigma$ and $<5\sigma$; $\vec{V}_{out}$). The term $\rho\left(\vec{V}_{in}, \vec{V}_{out}\right)$ represents the extent to which orientations inside the cell's receptive field match those outside of the receptive field for a given image. In the case of the Iso stimulus, $\rho(\vec{V}_{in}, \vec{V}_{out}) > 0$, yielding suppression, while in the case of Cross stimulus, $\rho(\vec{V}_{in}, \vec{V}_{out}) < 0$, yielding facilitation. We set the scaling factor $\beta$ to 0.95 to ensure that the denominator never reaches zero.

## Neural network model

We analyzed decoding performance of intermediate layers of a deep network (VGG-16) trained for classification on ImageNet (**Simonyan and Zisserman, 2014**). We used the mean of the response magnitude to all 14 individual frames corresponding to a given stimulus to represent a given unit's response to a given trial type. We extracted this response for each of the five max-pool layers in the network and computed decoding curves using the same procedures as for the neural data.

## Noise factor

To account for neural response variability, we added noise to all models proportional to the mean of responses for the simulated network within each condition. To achieve this, we added zero-mean, normally distributed fluctuations to each trial with variance proportional to the mean response across neurons followed by linear rectification to ensure positive outputs. Thus, the final output of a population would be equal to $response + N\left(0, noise_{factor} * pop_{mean}\right)$. Sweeps across various noise factors are shown in **Figure 7—figure supplements 1 and 2**.

## Acknowledgements

This work was supported by NIH (DP1-NS083063) and the Howard Hughes Medical Institute. We thank Audo Flores and Daniel Wagenaar for technical support, Sotiris Masmanidis for supplying the silicon recording probes, and David Fitzpatrick and Yong-Gang Yau for invaluable help setting up a tree shrew colony. FJL was supported by an Arnold O Beckman postdoctoral fellowship and a Burroughs Wellcome PDEP Award. DH and CLAH were supported by the Swiss National Science Foundation (310030E_190060) and the Human Frontiers Science Program (RGP0024/2016).

## Additional information

### Funding

| Funder | Grant reference number | Author |
|---|---|---|
| National Institutes of Health | DP1-NS083063 | Doris Y Tsao |
| Arnold O. Beckman postdoctoral fellowship | | Francisco J Luongo |
| Swiss National Science Foundation | 310030E_190060 | Daniel Huber |
| Howard Hughes Medical Institute | | Doris Y Tsao |
| Burroughs Wellcome PDEP Award | | Francisco J Luongo |

| Funder | Grant reference number | Author |
|---|---|---|
| Human Frontier Science Program | RGP0024/2016 | Daniel Huber |

The funders had no role in study design, data collection, and interpretation, or the decision to submit the work for publication.

## Author contributions

Francisco J Luongo, Conceptualization, Data curation, Software, Formal analysis, Funding acquisition, Investigation, Visualization, Methodology, Writing – original draft, Writing – review and editing, collected all mouse data together with L.L.He also collected treeshrew data together with JBW and FL; Lu Liu, Conceptualization, Data curation, Software, Formal analysis, Investigation, Visualization, Methodology, Writing – original draft, Writing – review and editing; Chun Lum Andy Ho, Formal analysis, Investigation, Methodology, Writing – review and editing, collected mouse lemur data under supervision of DH; Janis K Hesse, Formal analysis, Investigation, Methodology, Writing – review and editing, collected macaque data; Joseph B Wekselblatt, Formal analysis, Investigation, Methodology, Writing – review and editing, collected treeshrew data together with FL and FJL; Frank F Lanfranchi, Formal analysis, Investigation, Methodology, Writing – review and editing, collected treeshrew data together with JBW and FJL; Daniel Huber, Resources, Supervision, Funding acquisition, Project administration, Writing – review and editing, supervised collection of mouse lemur data; Doris Y Tsao, Conceptualization, Resources, Supervision, Funding acquisition, Visualization, Methodology, Writing – original draft, Project administration, Writing – review and editing, collected all mouse data together with FJL

## Author ORCIDs

Lu Liu ![ORCID] http://orcid.org/0009-0004-9272-5996
Doris Y Tsao ![ORCID] http://orcid.org/0000-0003-1083-1919

## Ethics

The following animals were used in this study: adult mice 2–12 months old, both male and female; adult treeshrews 7–18 months old, both male and female; adult mouse lemurs 2–3.5 years, both male and female; and adult macaques 3 and 7 years old, male. All procedures on mice, macaques, and treeshrews were conducted in accordance with the ethical guidelines of the National Institutes of Health and were approved by the Institutional Animal Care and Use Committee at the California Institute of Technology. Mouse lemur experiments were in accordance with European animal welfare regulations and were reviewed by the local ethics committee (Comite d'éthique en expérimentation animale No. 68) in Brunoy, France, by the ethics committee of the University of Geneva, Switzerland and authorized by the French 'Ministére de l'education nationale de l'enseignement supérieur et de la recherche'.

## Decision letter and Author response

Decision letter https://doi.org/10.7554/eLife.74394.sa1
Author response https://doi.org/10.7554/eLife.74394.sa2

---

# Additional files

## Supplementary files

• Transparent reporting form

## Data availability

Source data have been provided to replicate all neural and behavioral figures (2,3,4,5,6,7). These data have been uploaded to dryad: https://doi.org/10.5061/dryad.ngf1vhhvp. Sufficient modeling details have been provided in methods section to replicate relevant parts of figure 8.

The following dataset was generated:

| Author(s) | Year | Dataset title | Dataset URL | Database and Identifier |
|---|---|---|---|---|
| Tsao DY | 2023 | Data from: Mice and primates use distinct strategies for visual segmentation | https://dx,doi.org/10.5061/dryad.ngf1vhhvp | Dryad Digital Repository, 10.5061/dryad.ngf1vhhvp |

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
