## [Editor Report]

There is abundant evidence for differences in the organization of the visual system between primates and rodents. How do these differences yield distinct behaviors in these species? The authors show a major difference in the ability of mice and primates in detecting figures from ground based on motion and texture patterns, revealing a fundamental limitations of mice in segmenting visual scenes.

---

## [Decision Letter]

**Decision letter after peer review:**

Thank you for submitting your article "Mice and primates use distinct strategies for visual segmentation" for consideration by *eLife*. Your article has been reviewed by 2 peer reviewers, and the evaluation has been overseen by a Reviewing Editor and Joshua Gold as the Senior Editor. The following individual involved in review of your submission has agreed to reveal their identity: Matthew W. Self (Reviewer #1).

The reviewers have discussed their reviews with one another, and the Reviewing Editor has drafted this to help you prepare a revised submission. Overall, the reviewers were positive and we can recommend publication, provided you can fulfill a few requests for additional analyses.

Essential revisions:

1) The data would be more convincing if the authors could show that indeed activity changes of the neurons are really due to different stimuli and not due to activity changes with time or brain state. We recommend that at the very least the cell should be visually responsive throughout the 6 blocks that go into the analyses for figures 4, 5 and 6.

2) Figure 1c seems not very relevant for this paper and is hard to interpret without additional explanation.

3) In general, the reporting in the text and figures, as well as the methods is incomplete. Neither for electrophysiology, nor imaging, the number of cells going into the analyses is reported, nor are details on spike sorting (are all units single units? What are the criteria for cell inclusion?) or laminar location of their neurons for electrophysiology.

4) Similarly, reporting on the RFs (sizes, percentages of cells with significant fits, differences among areas…), as well as details on the mapping for example for the sparse noise (size of the pixels, sampled area, timing, analysis window…) are completely missing. The schematized RFs in Figure 3 c and d seem misleadingly small. I am assuming the green ellipse in 4a is an actually measured RF? This should be stated in the legend.

5) It is only mentioned for imaging that there were 7 blocks of data collection (6 conditions plus RF mapping). Were these always presented in the same order or randomized? Was it the same for imaging and ephys?

6) We recommend that Figure 3f reports the correlation values reported so the reader can judge how representative the example neurons are. The spatial masks are not described in the text, legend or methods. Are they the mapped receptive fields? Why are they on a square and not on the 16x8 grid?

7) Line 324 states that within conditions the distributions were not different from 0, but the statistics describe areas, not conditions. At least in LM it looks like the cross population could well be off 0. The difference from 0 should be tested for all 9 distributions and reported. Figures 5 d,e and g,h could be performed separately for each area, as is done in the supplemental figure for the PSTHs. But at least the fraction of cells from each area going into the batched analysis should be stated.

8) Schnabel et al. 2020 show even more robust modulation by a figure edge than figure center in mouse V1. It could be tested, if this result holds in this dataset as well.

---

## [Author Response]

Essential Revisions (for the authors):1) The data would be more convincing if the authors could show that indeed activity changes of the neurons are really due to different stimuli and not due to activity changes with time or brain state. We recommend that at the very least the cell should be visually responsive throughout the 6 blocks that go into the analyses for figures 4, 5 and 6.

The three different types of stimuli (Cross/Iso/Nat) were presented *randomly interleaved*, and within each block the two stimulus conditions were further randomly interleaved. In a typical experiment, we would run 10 trials/block, and ~6 blocks/experiment (thus each of the three types of stimuli, Cross/Iso/Nat, were repeated twice). Therefore, it seems unlikely that differences observed between conditions arose due to brain state changes or other changes over time.

Furthermore, note that we only chose visually-responsive cells which had clear receptive fields for the analysis in Figure 5. Specifically, we state in the Methods:

“[W]e used 2D Gaussian to fit the RFs. The cells with RF’s goodness of fit (GOF) above 0.1 were included for subsequent analyses of electrophysiological data (Figures 4, 5, 6, Figure 5—figure supplement 1).”

In one experiment, we mapped the RFs before and after the Cross/Iso/Nat stimuli. As shown in Author response image 1, the RFs of all the cells recorded in this session (16 cells) had similar structure before and after the Cross/Iso/Nat experiment. Again, this shows that the activity changes of neurons were unlikely to have been due to brain state changes affecting visual responsiveness.

**Author response image 1. sa2fig1:** Receptive field maps of 16 cells recorded before and after mapping responses to the Cross/Iso/Nat experiment of Figures 4-6 in one session. Red = ON responses, Green = OFF responses.

2) Figure 1c seems not very relevant for this paper and is hard to interpret without additional explanation.

We have now deleted this figure.

3) In general, the reporting in the text and figures, as well as the methods is incomplete. Neither for electrophysiology, nor imaging, the number of cells going into the analyses is reported, nor are details on spike sorting (are all units single units? What are the criteria for cell inclusion?) or laminar location of their neurons for electrophysiology.

Thank you for pointing this out. We have added the following details related to electrophysiology to the Methods:

“We used Kilosort (Pachitariu et al., 2016) for spike sorting of the data. The output of the automatic template-matching algorithm from Kilosort was visualized on Phy and then curated manually. Mixed single units and multi-units were included. We have two criteria for the cells included in our population analysis: (1) RF goodness of fit (GOF) > 0.1; (2) Total spikes > 100 for each of the six stimulus blocks.”

We did not track the laminar location of neurons for electrophysiology.

We have added the # cells used for each analysis (both electrophysiology and imaging) into the appropriate figure legends.

4) Similarly, reporting on the RFs (sizes, percentages of cells with significant fits, differences among areas…), as well as details on the mapping for example for the sparse noise (size of the pixels, sampled area, timing, analysis window…) are completely missing. The schematized RFs in Figure 3 c and d seem misleadingly small. I am assuming the green ellipse in 4a is an actually measured RF? This should be stated in the legend.

We computed the RF sizes based on Gaussian fit. The size for each RF was computed as the mean of the standard deviation along the x and y directions in the Gaussian fit. Only cells with GOF > 0.1 were included in our population analyses (Figures 4, 5, 6, Figure 5—figure supplement 1). The percentages of cells with significant fits among the three areas was: V1: 67%; LM: 66%; RL: 51%.

**Author response image 2. sa2fig2:** Distributions of RF sizes recorded in all areas (top), and in V1, RL, and LM individually (bottom).

We have now added the following details on RF mapping with electrophysiology to the Methods:

“For electrophysiology, we used a sparse noise stimulus to map the spatial RFs in all three areas. We divided the screen into a 18x32 grid of squares, each 3.5° on a side (each grid square was 2.2 x 2.2 cm, 18 cm from the mouse’s left eye). In each stimulus frame, one grid square, black or white, was presented alternately. The stimulus contained 8000 frames in total and was presented at 10 Hz. RFs were computed by spike triggered average (STA). The analysis window for STA was 30-100 ms. Then we used 2D Gaussian to fit the RFs. The cells with RF goodness of fit (GOF) above 0.1 were include in subsequent analyses. We limited our analysis for figure-ground and border-ownership modulation in Figure 5 to neurons that had significant receptive field fits and used the central position of that fit as the neuron’s inferred RF center. For population neural decoding analyses, we used all neurons regardless of receptive field fits.”

In addition, we include the following text in the Methods with details on RF mapping using two photon imaging:

“*RF mapping.* Receptive fields were mapped for neurons under 2-photon using an isolated drifting Gabor patch stimulus: a patch of ~6° containing a drifting Gabor appeared in one of three different orientations (45°, 180°, 315°) and 2 directions at a random position. We repeated this procedure for 9 repeats and at 16 by 9 positions and then collapsed responses across all trial types to compute a spatial PSTH. We then fit a 2D Gaussian to the response and classified neurons as having a significant receptive field fit if the goodness of fit exceeded the goodness of fit for at least 99 out of 100 bootstrapped trials where spatial location was shuffled across trials.”

The green ellipse in Figure 3 is purely schematic, and it not taken from real data. In Figure 4, the green ellipses are fits from real data.

5) It is only mentioned for imaging that there were 7 blocks of data collection (6 conditions plus RF mapping). Were these always presented in the same order or randomized? Was it the same for imaging and ephys?

We now clarify in the methods that for both electrophysiology and two-photon imaging experiments, the 6 conditions were randomly interleaved. RF mapping was performed at the beginning of the session, and sometimes also at the end (e.g., see Author response image 1):

“For both electrophysiology and two-photon imaging experiments, the 6 conditions (Cross, Iso, Nat x 2 conditions) were pseudo-randomly interleaved, i.e., Cross/Iso/Nat were presented in a random order, and the two within-condition repetitions (e.g., two orientations or textures) were presented consecutively but also randomized.”

6) We recommend that Figure 3f reports the correlation values reported so the reader can judge how representative the example neurons are. The spatial masks are not described in the text, legend or methods. Are they the mapped receptive fields? Why are they on a square and not on the 16x8 grid?

We have now added the correlation values for the examples in Figure 3f to the figure legend.

The following clarification regarding the spatial mask has been added to the figure legend for Figure 3f:

“Spatial masks are from suite2P spatial filters and are meant to illustrate qualitatively similar morphology in matched neurons across days.”

7) Line 324 states that within conditions the distributions were not different from 0, but the statistics describe areas, not conditions. At least in LM it looks like the cross population could well be off 0. The difference from 0 should be tested for all 9 distributions and reported. Figures 5 d,e and g,h could be performed separately for each area, as is done in the supplemental figure for the PSTHs. But at least the fraction of cells from each area going into the batched analysis should be stated.

We have performed the t-test for the 9 distributions from 0, the mean and P values have now added this to the figure legend for Figure 5b.

The percentages of cells with significant fits and included in the analysis among the three areas was: V1: 67%; LM: 66%; RL: 51%.

8) Schnabel et al. 2020 show even more robust modulation by a figure edge than figure center in mouse V1. It could be tested, if this result holds in this dataset as well.

Consistent with Schnabel et al. 2020, we found that most cells in V1 respond more strongly to edges than figure for both cross and iso conditions.

**Author response image 3. sa2fig3:**